# Q-Distribution guided Q-learning for offline reinforcement learning: Uncertainty penalized Q-value via consistency model

**Jing Zhang**
HKUST
jzhanggy@connect.ust.hk

**Linjiajie Fang**
HKUST
lfangad@connect.ust.hk

**Kexin Shi**
HKUST
kshiaf@connect.ust.hk

**Wenjia Wang**[*]
HKUST (GZ)
wenjiawang@hkust-gz.edu.cn

**Bing-Yi Jing**[*]
SUSTech
jingby@sustech.edu.cn

## Abstract

"Distribution shift" is the main obstacle to the success of offline reinforcement learning. A learning policy may take actions beyond the behavior policy's knowledge, referred to as Out-of-Distribution (OOD) actions. The Q-values for these OOD actions can be easily overestimated. As a result, the learning policy is biased by using incorrect Q-value estimates. One common approach to avoid Q-value overestimation is to make a pessimistic adjustment. Our key idea is to penalize the Q-values of OOD actions associated with high uncertainty. In this work, we propose Q-Distribution Guided Q-Learning (QDQ), which applies a pessimistic adjustment to Q-values in OOD regions based on uncertainty estimation. This uncertainty measure relies on the conditional Q-value distribution, learned through a high-fidelity and efficient consistency model. Additionally, to prevent overly conservative estimates, we introduce an uncertainty-aware optimization objective for updating the Q-value function. The proposed QDQ demonstrates solid theoretical guarantees for the accuracy of Q-value distribution learning and uncertainty measurement, as well as the performance of the learning policy. QDQ consistently shows strong performance on the D4RL benchmark and achieves significant improvements across many tasks.

## 1 Introduction

Reinforcement learning (RL) has seen remarkable success by using expressive deep neural networks to estimate the value function or policy function [1]. However, in deep RL optimization, updating the Q-value function or policy value function can be unstable and introduce significant bias [2]. Since the learning policy is influenced by the Q-value function, any bias in the Q-values affects the learning policy. In online RL, the agent's interaction with the environment helps mitigate this bias through reward feedback for biased actions. However, in offline RL, the learning relies solely on data from a behavior policy, making information about rewards for states and actions outside the dataset's distribution unavailable.

---

[*]Corresponding authors: Bing-Yi Jing and Wenjia Wang.

38th Conference on Neural Information Processing Systems (NeurIPS 2024).

It is commonly observed that during offline RL training, backups using OOD actions often lead to target Q-values being *overestimated* [3] (see Figure 1(a)). As a result, the learning policy tends to prioritize these risky actions during policy improvement. This false prioritization accumulates with each training step, ultimately leading to failure in the offline training process [4, 5, 6]. Therefore, addressing Q-value overestimation for OOD actions is crucial for the effective implementation of offline reinforcement learning.

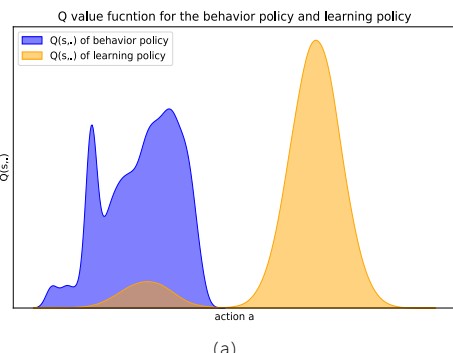 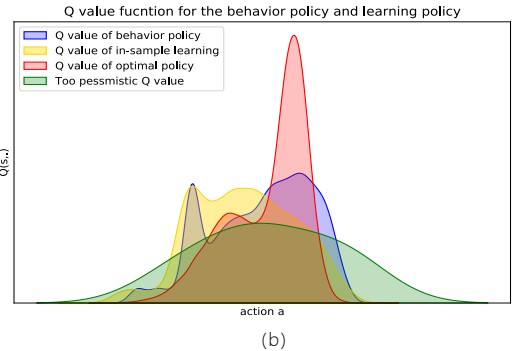

Figure 1: (a) The maximum of the estimated Q-value often occurs in OOD actions due to the instability of the offline RL backup process and the "distribution shift" problem , so the Q-value of the learning policy (yellow line) will diverge from the behavior policy's action space (blue line) during the training. (b) The red line represents the optimal Q-value within the action space of the dataset, while the blue line depicts the Q-value function of the behavior policy. The gold line corresponds to the Q-value derived from the in-sample Q training algorithm, showcasing a distribution constrained by the behavior policy. On the other hand, the green line illustrates the Q-value resulting from a more conservative Q training process. Although it adopts lower values in OOD actions, the Q-value within in-distribution areas proves excessively pessimistic, failing to approach the optimal Q-value.

Since any bias or error in the Q-value will propagate to the learning policy, it's crucial to evaluate whether the Q-value is assigned to OOD actions and to apply a pessimistic adjustment to address overestimation. Ideally, this adjustment should only target OOD actions. One common way to identify whether the Q-value function is updated by OOD actions is by estimating the uncertainty of the Q-value [4] in the action space. However, estimating uncertainty presents significant challenges, especially with high-capacity Q-value function approximators like neural networks [4]. If Q-value uncertainty is not accurately estimated, a penalty may be uniformly applied across most actions [7], hindering the optimality of the Q-value function.

While various methods [7, 8, 9, 10, 11, 12, 3, 13, 14] attempt to make pessimistic estimates of the Q-value function, most have not effectively determined which Q-values need constraining or how to pessimistically estimate them with reliable and efficient uncertainty estimates. As a result, previous methods often end up being overly conservative in their Q-value estimations [15] or fail to achieve a tight lower confidence bound of the optimal Q-value function. Moreover, some in-sample training [16, 17, 18, 19] of the Q-value function may lead it to closely mimic the Q-value of the behavior policy (see Figure 1(b)), rendering it unable to surpass the performance of the behavior policy, especially when the behavior policy is sub-optimal. Therefore, in balancing Q safety for learning and not hindering the recovery of the most optimal Q-value, current methods tend to prioritize safe optimization of the Q-value function.

In this study, we introduce Q-Distribution guided Q-Learning (QDQ) for offline RL [2]. The core concept focuses on estimating Q-value uncertainty by directly computing this uncertainty through bootstrap sampling from the behavior policy's Q-value distribution. By approximating the behavior policy's Q-values using the dataset, we train a high-fidelity and efficient distribution learner-consistency model [20]. This ensures the quality of the learned Q-value distribution.

---

[2]Our code can be found at https://github.com/evalarzj/qdq.

Since the behavior and learning policies share the same set of high-uncertainty actions [5], we can sample from the learned Q-value distribution to estimate uncertainty, identify risky actions, and make the Q target values for these actions more pessimistic. We then create an uncertainty-aware optimization objective to carefully penalize Q-values that may be OOD, ensuring that the constraints are appropriately pessimistic without hindering the Q-value function's exploration in the in-distribution region. QDQ aims to find the optimal Q-value that exceeds the behavior policy's optimal Q-value while remaining as pessimistic as possible in the OOD region. Moreover, our pessimistic approach is robust against errors in uncertainty estimation. Our main contributions are as follows:

- **Utilization of trajectory-level data with a sliding window**: We use trajectory-level data with a sliding window approach to create the real truncated Q dataset. Our theoretical analysis (Theorem 4.1) confirms that the generated data has a distribution similar to true Q-values. Additionally, distributions learned from this dataset tend to favor high-reward actions.

- **Introduction of a consistency model as a distribution learner**: QDQ introduces the consistency model [20] as the distribution learner for the Q-value. Similar to the diffusion model, the consistency model demonstrates strong capabilities in distribution learning. Our theoretical analysis (Theorem 4.2) highlights its consistency and one-step sampling properties, making it an ideal choice for uncertainty estimation.

- **Risk estimation of Q-values through uncertainty assessment**: QDQ estimates the risk set of Q-values by evaluating the uncertainty of actions. For Q-values likely to be overestimated and associated with high uncertainty, a pessimistic penalty is applied. For safer Q-values, a mild adjustment based on uncertainty error enhances their robustness.

- **Uncertainty-aware optimization objective to address conservatism**: To reduce the overly conservative nature of pessimistic Q-learning in offline RL, QDQ introduces an uncertainty-aware optimization objective. This involves simultaneous optimistic and pessimistic learning of the Q-value. Theoretical (Theorem 4.3 and Theorem 4.4) and experimental analyses show that this approach effectively mitigates conservatism issues.

## 2 Background

Our approach aims to temper the Q-values in OOD areas to mitigate the risk of unpredictable extrapolation errors, leveraging uncertainty estimation. We estimate the uncertainty of Q-values across actions visited by the learning policy using samples from a learned conditional Q-distribution via the consistency model. In this section, we provide a concise overview of the problem settings in offline RL and introduce the consistency model.

### 2.1 Fundamentals in offline RL

The online RL process is shaped by an infinite-horizon Markov decision process (MDP): $\mathcal{M} = \{\mathcal{S}, \mathcal{A}, \mathbb{P}, r, \mu_0, \gamma\}$. The state space is $\mathcal{S}$, and $\mathcal{A}$ is the action space. The transition dynamic among the state is determined by $\mathbb{P} : \mathcal{S} \times \mathcal{A} \mapsto \Delta(\mathcal{S})$, where $\Delta(\mathcal{S})$ is the support of $\mathcal{S}$. The reward determined on the whole state and action space is $r : \mathcal{S} \times \mathcal{A} \mapsto \mathbb{R}, r < \infty$, and can either be deterministic or random. $\mu_0(s_0)$ is the distribution of the initial states $s_0$, $\gamma \in (0, 1)$ is the discount factor. The goal of RL is to find the optimal policy $\pi : \mathcal{S} \mapsto \Delta(a)$ that yields the highest long-term average return:

$$J(\pi) = \mathbb{E}_{s_0 \sim \mu_0} V(s_0) = \mathbb{E}_{s_0 \sim \mu_0, a_0 \sim \pi} Q(s_0, a_0) = \mathbb{E}_{s_0 \sim \mu_0, a_0 \sim \pi} \left[ \mathbb{E}_\pi \left[ \sum_{k=1}^{\infty} \gamma^{k-1} r_k | s_k, a_k \right] \right], \quad (1)$$

where $Q(s, a)$ is the Q-value function under policy $\pi$. The process of obtaining the optimal policy is generally to recover the optimal Q-value function, which maximizes the Q-value function over the whole space $\mathcal{A}$, and then to obtain either an implicit policy (Q-Learning algorithm [21, 22, 23]), or a parameterized policy (Actor-Critic algorithm [24, 25, 26, 27, 28]).

The optimal Q-value function $Q^*(s, a)$ can be obtained by minimizing the Bellman residual:

$$\mathbb{E}_{s \sim \mathbb{P}, a \sim \pi} [Q(s, a) - \mathcal{B}Q(s, a)]^2, \quad (2)$$

where $\mathcal{B}$ is the Bellman operator defined as

$$\mathcal{B}Q(s,a) := r(s,a) + \gamma \mathbb{E}_{s' \sim \mathbb{P}}[\max_{a' \sim \pi} Q(s',a')].$$

However, the whole paradigm needs to be adjusted in the offline RL setting, as MDP is only determined from a dataset $\mathcal{D}$, which is generated by behavior policy $\pi_\beta$. Hence, the state and action space is constraint by the distribution support of $\mathcal{D}$. We redefine the MDP in the offline RL setting as: $\mathcal{M}_\mathcal{D} = \{\mathcal{S}_\mathcal{D}, \mathcal{A}_\mathcal{D}, \mathbb{P}_\mathcal{D}, r, \mu_0, \gamma\}$, where $\mathcal{S}_\mathcal{D} = \{s | s \in \Delta(s^\mathcal{D})\}$, $\mathcal{A}_\mathcal{D} = \{a | a \in \Delta(\pi_\beta)\}$. Then the transition dynamic is determined by $\mathbb{P}_D : \mathcal{S}_\mathcal{D} \times \mathcal{A}_\mathcal{D} \mapsto \Delta(\mathcal{S}_\mathcal{D})$. Therefore, the well-known "distribution shift" problem occurs when solving the Bellman equation Eq.2. The Bellman residual is taking expectation in $\mathcal{S}_\mathcal{D} \times \mathcal{A}_\mathcal{D}$, while the target Q-value is calculated based on the actions from the learning policy.

## 2.2 Consistency model

The consistency model is an enhanced generative model compared to the diffusion model. The diffusion model gradually adds noise to transform the target distribution into a Gaussian distribution and by estimating the random noise to achieve the reverse process, i.e., sampling a priori sample from a Gaussian distribution and denoise to the target sample iteratively (forms the sample generation trajectory). The consistency model is proposed to ensures each step in a sample generation trajectory of the diffusion process aligns with the target sample (we call consistency). Specifically, the consistency model [20] try to overcome the slow generation and inconsistency over sampling trajectory generated by the Probability Flow (PF) ODE during training process of the diffusion model [29, 30, 31, 32, 33].

Let $p_{data}(\boldsymbol{x})$ denote the data distribution, we start by diffuse the original data distribution by the PF ODE:

$$d\boldsymbol{x_t} = \left[\boldsymbol{\mu}(\boldsymbol{x_t}, t) - \frac{1}{2}\sigma(t)^2 \nabla \log(p_t(\boldsymbol{x_t}))\right] dt, \tag{3}$$

where $\boldsymbol{\mu}(\cdot, \cdot)$ is the drift coefficient, $\sigma(\cdot)$ is the diffusion coefficient, $p_t(\boldsymbol{x_t})$ is the distribution of $\boldsymbol{x_t}$, $p_0(\boldsymbol{x}) \equiv p_{data}(\boldsymbol{x})$, and $\{\boldsymbol{x_t}, t \in [\epsilon, T]\}$ is the solution trajectory of the above PF ODE.

Consistency model aims to learn a consistency function $f_\theta(\boldsymbol{x_t}, t)$ that maps each point in the same PF ODE trajectory to its start point, i.e., $f_\theta(\boldsymbol{x_t}, t) = \boldsymbol{x_\epsilon}, \forall t \in [\epsilon, T]$. Therefore, $\forall t, t' \in [\epsilon, T]$, we have $f_\theta(\boldsymbol{x_t}, t) = f_\theta(\boldsymbol{x_{t'}}, t')$, which is the "self-consistency" property of consistency model.

Here, $f_\theta(\boldsymbol{x_t}, t)$ is defined as:

$$f_\theta(\boldsymbol{x_t}, t) = c_{skip}(t)\boldsymbol{x_t} + c_{out}(t)F_\theta(\boldsymbol{x_t}, t), \tag{4}$$

where $c_{skip}(t)$ and $c_{out}(t)$ are differentiable functions, and $c_{skip}(\epsilon) = 1, c_{out}(\epsilon) = 0$ such that they satisfy the boundary condition $f_\theta(\boldsymbol{x_\epsilon}, \epsilon) = x_\epsilon$. In (4), $F_\theta(\boldsymbol{x_t}, t)$ can be free-form deep neural network with output that has the same dimension as $\boldsymbol{x_t}$.

Consistency function $f_\theta(\boldsymbol{x_t}, t)$ can be optimized by minimizing the difference of points in the same PF ODE trajectory. If we use a pretrained diffusion model to generate such PF ODE trajectory, then utilize it to train a consistency model, this process is called consistency distillation. We use the consistency distillation method to learn a consistency model in this work and optimize the consistency distillation loss as the Definition 1 in [20].

With a well-trained consistency model $f_\theta(\boldsymbol{x_t}, t)$, we can generate samples by sampling from the initial Gaussian distribution $\hat{\boldsymbol{x}}_T \sim \mathcal{N}(\boldsymbol{0}, T^2\boldsymbol{I})$ and then evaluating the consistency model for $\hat{\boldsymbol{x}}_\epsilon = f_\theta(\hat{\boldsymbol{x}}_T, T)$. This involves only one forward pass through the consistency model and therefore generates samples in a single step. This is the one-step sampling process of the consistency model.

## 3 Q-Distribution guided Q-learning via Consistency Model

In this work, we present a novel method for offline RL called Q-Distribution Guided Q-Learning (QDQ). First, we quantify the uncertainty of Q-values by learning the Q-distribution using the consistency model. Next, we propose a strategy to identify risky actions and penalize their Q-values based on uncertainty estimation, helping to mitigate the associated risks. To tackle the excessive conservatism seen in previous approaches, we introduce uncertainty-aware Q optimization within the Actor-Critic learning framework. This mechanism allows the Q-value function to perform both optimistic and pessimistic optimization, fostering a balanced approach to learning.

### 3.1 Learn Uncertainty of Q-value by Q-distribution

Estimating the uncertainty of the Q function is a significant challenge, especially with deep neural network Q estimators. A practical indicator of uncertainty is the presence of large variances in the estimates. Techniques such as bootstrapping multiple Q-values and estimating variance [7] have been used to address this issue. However, these ensemble methods often lack diversity in Q-values [9] and fail to accurately represent the true Q-value distribution. They may require tens or hundreds of Q-values to improve accuracy, which is computationally inefficient [9, 5].

Other approaches involve estimating the Q-value distribution and determining the lower confidence bound [10, 12, 16], or engaging in in-distribution learning of the Q-value function [3, 13, 16, 10, 34]. However, these methods often struggle to provide precise uncertainty estimations for the Q-value [15]. Stabilization methods can still lead to Q-value overestimation [3], while inaccurate variance estimation can worsen this problem. Furthermore, even if the Q-value is not overestimated, there is still a risk of it being overly pessimistic or constrained by the performance of the behavior policy when using in-distribution-only training.

In this subsection, we elucidate the process of learning the distribution of Q-values based on the consistency model, and outline the technique for estimating the uncertainty of actions and identifying risky actions. We have give a further demonstration on the performance of the consistency model and efficiency of the uncertainty estimation in Appendix G.2 and Appendix G.3.

**Trajectory-level truncated Q-value.** We chose to estimate the Q-value distribution of the behavior policy instead of the learning policy because they share a similar set of high-uncertainty actions [5]. Using the behavior policy's Q-value distribution has several advantages. First, the behavior policy's Q-value dataset comes from the true dataset, ensuring high-quality distribution learning. In contrast, the learning policy's Q-value is unknown, counterfactually learned, and often noisy and biased, leading to poor data quality and biased distribution learning. Second, using the behavior policy's Q-value distribution to identify high-uncertainty actions does not force the learning policy's target Q-value to align with that of the behavior policy.

To gain insights into the Q-value distribution of the behavior policy, we first need the raw Q-value data. The calculation of the Q-value operates at the trajectory level, represented as $\tau = (s_0, a_0, s_1, a_1, ...)$, with an infinite horizon (see Eq.1). In the context of offline RL, our training relies on the dataset $\mathcal{D}$ produced by the behavior policy. This dataset consists of trajectories generated by the behavior policy, which is the only available trajectory-level data. However, the trajectory-level data from the behavior policy often faces a significant challenge: sparsity. This issue becomes even more pronounced when dealing with low-quality behavior policies, as the generated trajectories tend to be sporadic and do not adequately cover the entire state-action space $\mathcal{S} \times \mathcal{A}$, especially the high reward region.

To address this pervasive issue of sparsity, as well as the infinite summation in Eq.1, we present a novel approach aimed at enhancing sample efficiency. Our proposed solution involves the utilization of truncated trajectories to ameliorate the sparsity conundrum and avoid infinite summation. By employing a $k$- step sliding window of width $\mathcal{T}$, we systematically traverse the original trajectories, isolating segments within the window to compute the truncated Q-value (as depicted in Figure A.1). For instance, considering the initiation point of the $i$-th step sliding window as $(s_i, a_i)$, by setting $\mathcal{T} = i + k$, we derive the truncated Q-value of this starting point as follows:

$$Q_{\mathcal{T}}^{\pi_\beta}(s_i, a_i) = \sum_{m=i}^{\mathcal{T}} \gamma^{m-1} r(s_m, a_m) \times t(s_m, a_m), t(s_m, a_m) = \begin{cases} 0, & terminal, \\ 1, & otherwise. \end{cases} \quad (5)$$

The truncation of Q-values can occur either through sliding window mechanisms or task terminations. When truncation happens due to termination, the Q-value from Eq.5 is equivalent to the true Q-value, $Q^{\pi_\beta}\mathcal{T}(\cdot, \cdot) \equiv Q^{\pi_\beta}(\cdot, \cdot)$. In contrast, if truncation results from window blocking, our theoretical analysis in Theorem 4.1 confirms that the distribution of truncated Q-values has properties similar to those of the true Q-value distribution.

Using a $k$-step sliding window does not compromise the consistency of the trajectory, owing to the inherent memory-less Markov property in RL. This strategic truncation allows for the extraction of truncated Q-values, which can improve sample efficiency, especially for long trajectories. Moreover, this approach highlights actions with potential high Q-values, as actions from lengthy trajectories—those with many successful interactions—are encountered more often during Q-distribution

training. Consequently, the uncertainty of these actions is lower, reducing the likelihood of them being overly pessimistic.

**Learn the distribution of Q-value.** In distributional RL, the learning of Q-value distributions is typically achieved through Gaussian neural networks [35, 36], Gaussian processes [37, 38], or categorical parameterization [39]. However, these methods often suffer from low precision representation of Q-value distributions, particularly in high-dimensional spaces. Moreover, straightforward replacement of true Q-value distributions with ensembles or bootstraps can lead to reduced accuracy in uncertainty estimation(a critical aspect in offline reinforcement learning [4]), or impose significant computational burdens [8, 7].

The idea of diffusing the original distribution using random noise has rendered the diffusion model a potent and high-fidelity distribution learner. However, it has limitations when estimating uncertainty. Sampling with a diffusion model requires a multi-step forward diffusion process to ensure sample quality. Unfortunately, this iterative process can compromise the accuracy of uncertainty estimates by introducing significant fluctuations and noise into the Q-value uncertainty. For a detailed discussion, see Appendix A.2.

To address this issue, we suggest using the consistency model [20] to learn the Q-value distribution. The consistency model allows for one-step sampling, like other generative models, which reduces the randomness found in the multi-step sampling of diffusion models. This results in a more robust uncertainty estimation. Furthermore, the consistency feature, as explained in Theorem 4.2, accurately captures how changes in actions affect the variance of the final bootstrap samples, making Q-value uncertainty more sensitive to out-of-distribution (OOD) actions compared to the diffusion model. Additionally, the fast-sampling process of the consistency model improves QDQ's efficiency. While there may be some quality loss in restoring real samples, this is negligible for QDQ since it only calculates uncertainty based on the variance of the bootstrap samples, not the absolute Q-value of the sampled samples. Overall, the consistency model is an ideal distribution learner for uncertainty estimation due to its reliability, high-fidelity, ease of training, and faster sampling.

Once we derive the truncated Q dataset $\mathcal{D}_Q$, we train a conditional consistency model, denoted by $f_\theta(x_T, T|(s, a))$, which approximates the distribution of Q-values. Since the consistency model aligns with one-step sampling, we can easily sample multiple Q-values for each action using the consistency model. Suppose we draw $n$ prior noise $\{\hat{x}_{T_1}, \hat{x}_{T_2}, \cdots, \hat{x}_{T_n}\}$ from the initial noise distribution $\mathcal{N}(0, T^2)$, and denoise the prior samples by the consistency one-step forward process: $\hat{x}_{\epsilon_i} = f_\theta(\hat{x}_{T_i}, T_i|(s, a)), i = 1, 2, \cdots, n$. Then the variance of these Q-values, derived by

$$V(X_\epsilon|(s, a)) = \frac{1}{n-1} \sum_{i=1}^{n} \left[ f_\theta(\hat{x}_{T_i}, T|(s, a)) - \frac{1}{n} \sum_{i=1}^{n} f_\theta(\hat{x}_{T_i}, T|(s, a)) \right]^2, \qquad (6)$$

can be used to gauge the uncertainty of $Q(s, a)$.

## 3.2 Q-distribution guided optimization in offline RL

**Recover Q-value function.** We propose an uncertainty-aware optimization objective $\mathcal{L}_{uw}(Q)$ to penalize Q-value for OOD actions as well as to avoid too conservative Q-value learning for in-distribution areas. The uncertainty-aware learning objective for Q-value function $Q_\theta(s, a)$ is :

$$\mathcal{L}_{uw}(Q_\theta) = \min_\theta \{\alpha \mathcal{L}(Q_\theta)_H + (1 - \alpha)\mathcal{L}(Q_\theta)_L\}. \qquad (7)$$

In Eq.7, $\mathcal{L}(Q_\theta)H$ represents the classic Bellman residual defined in Eq.2. This residual is used in online RL and encourages optimistic optimization of the Q-value. In contrast, $\mathcal{L}(Q\theta)_L$ is a pessimistic Bellman residual based on the uncertainty-penalized Q target $Q_L(s', a')$, defined as

$$Q_L(s', a') = \frac{1}{\mathcal{H}_Q(a'|s')} Q_\theta(s', a') 1_{(a' \in \mathcal{U}(Q))} + \beta Q_\theta(s', a') 1_{(a' \notin \mathcal{U}(Q))}. \qquad (8)$$

In Eq.8, $\mathcal{H}_Q(a'|s') = \sqrt{V(X_\epsilon|(s', a'))}$ represents the uncertainty estimate of the Q-value for action $a'$. The set $\mathcal{U}(Q)$ includes actions that may be out-of-distribution (OOD). We use the upper $\beta$-quantile $\mathcal{H}_Q^\beta(a'|s')$ of the uncertainty estimate on actions taken by the learning policy as the threshold for forming $\mathcal{U}(Q)$. Additionally, we incorporate the quantile parameter $\beta$ as a robust weighting factor for

the unpenalized Q-target value. This helps control the estimation error of uncertainty and enhances the robustness of the learning objective. We can also set a free weighting factor, but we use $\beta$ to reduce the number of hyperparameters.

**Improve the learning policy.** The optimization of learning policy follows the classic online RL paradigm:

$$\mathcal{L}_\phi(\pi) = \max_\phi \; \left[\mathbb{E}_{s\sim\mathbb{P}_\mathcal{D}(s), a\sim\pi_\phi(\cdot|s)}[Q_\theta(s,a)] + \gamma\mathbb{E}_{a\sim\mathcal{D}}[\log\pi_\phi(a)]\right]. \tag{9}$$

In Eq.9, an entropy term is introduced to further stabilize the volatile learning process of Q-value function. For datasets with a wide distribution, we can simply set the penalization factor $\gamma$ to zero, which can further enhance performance. Furthermore, other policy learning objectives, such as the AWR policy objective [40], can also be flexibly used within the QDQ framework, especially for the goal conditioned task like Antmaze.

We outline the entire learning process of QDQ in Algorithm 1. In Section 4, Theorems 4.3 and 4.4 show that QDQ penalizes the OOD region based on uncertainty while ensuring that the Q-value function in the in-distribution region is close to the optimal Q-value. This alignment is the main goal of offline RL.

---

**Algorithm 1** Q-Distribution guided Q-learning (QDQ)

---
**Initialize:** target network update rate $\kappa$, uncertainty-aware learning hyperparameter $\alpha, \beta$, policy training hyperparameters $\gamma$. Consistency model $f_\eta$, Q networks $\{Q_{\theta_1}, Q_{\theta_2}\}$, actor $\pi_\phi$, target networks $\{Q_{\theta'_1}, Q_{\theta'_2}\}$, target actor $\pi_{\phi'}$.
**Q-distribution learning:**
Calculate Q dataset $\mathcal{D}_Q = \{Q^{\pi_\beta}_\mathcal{T}(s,a)\}$ scanning each trajectory $\tau \in \mathcal{D}$ by Eq.5.
**for** each gradient step **do**
    Sample minibatch of $Q^{\pi_\beta}_\mathcal{T}(s,a) \sim \mathcal{D}_Q$
    Update $\eta$ minimizing consistency distillation loss in Eq.(7) [20]
**end for**
**for** each gradient step **do**
    Sample mini-batch of transitions $(s, a, r, s') \sim \mathcal{D}$
    **Updating Q-function:**
    Update $\theta = (\theta_1, \theta_2)$ minimizing $\mathcal{L}_{uw}(Q_\theta)$ in Eq.7
    **Updating policy:**
    Update $\phi$ minimizing $\mathcal{L}_\phi(\pi)$ in Eq.9
    **Update Target Networks:**
    $\phi' \leftarrow \kappa\phi + (1-\kappa)\phi'; \theta'_i \leftarrow \kappa\theta_i + (1-\kappa)\theta'_i, i = 1, 2$
**end for**

---

## 4 Theoretical Analysis

In this section, we provide a theoretical analysis of QDQ. The first theorem states that if $\mathcal{T}$ is sufficiently large, the distribution of $Q^{\pi_\beta}\mathcal{T}$ does not significantly differ from the true distribution of $Q^{\pi_\beta}$. This shows that our sliding window-based truncated Q-value distribution converges to the true Q-value distribution, ensuring accurate uncertainty estimation. A detailed proof can be found in Appendix B.

**Theorem 4.1** (Informal). *Under some mildly condition, the truncated Q-value $Q^{\pi_\beta}_\mathcal{T}$ converge in-distribution to the true true Q-value $Q^{\pi_\beta}$.*

$$F_{Q^{\pi_\beta}_\mathcal{T}}(x) \to F_{Q^{\pi_\beta}}(x), \mathcal{T} \to +\infty. \tag{10}$$

In Theorem 4.2, we analyze why the consistency model is suitable for estimating uncertainty. Our analysis shows that Q-value uncertainty is more sensitive to actions. This sensitivity helps in detecting out-of-distribution (OOD) actions. A detailed statement of the theorem and its proof can be found in Appendix C.

**Theorem 4.2** (Informal). *Following the assumptions as in [20], $f_\theta(x, T|(s,a))$ is L-Lipschitz. We also assume the truncated Q-value is bounded by $\mathcal{H}$. The action $a$ broadly influences $V(X_\epsilon|(s,a))$ by:* $|\frac{\partial var(X_\epsilon)}{\partial a}| = O(L^2 T\sqrt{\log n})\mathbf{1}$.

In Theorem 4.3, we give theoretical analysis that the uncertainty-aware learning objective in Eq.7 can converge and the details can be found in Appendix D.

**Theorem 4.3** (Informal). *The Q-value function of QDQ can converge to a fixed point of the Bellman equation:* $Q(s, a) = \mathscr{F}Q(s, a)$, *where the Bellman operator* $\mathscr{F}Q(s, a)$ *is defined as:*

$$\mathscr{F}Q(s, a) := r(s, a) + \gamma \mathbb{E}_{s' \sim P_D(s')} \{ \max_{a'} [\alpha Q(s', a') + (1 - \alpha) Q_L(s', a')] \}. \tag{11}$$

Theorem 4.4 shows that QDQ penalizes the OOD region by uncertainty while ensuring that the Q-value function in the in-distribution region is close to the optimal Q-value, which is the goal of offline RL.

**Theorem 4.4** (Informal). *Under mild conditions, with probability* $1 - \eta$ *we have*

$$\left\| Q^\Delta - Q^* \right\|_\infty \leq \epsilon, \tag{12}$$

*where* $Q^\Delta$ *is learned by the uncertainty-aware loss in Eq.7,* $\epsilon$ *is error rate related to the difference between the classical Bellman operator* $\mathcal{B}Q$ *and the QDQ bellman operator* $\mathscr{F}Q$.

The optimal Q-value, $Q^\Delta$, derived by the QDQ algorithm can closely approximate the optimal Q-value function, $Q^*$, benefiting from the balanced approach of the QDQ algorithm that avoids excessive pessimism for in-distribution areas. Both the value $\epsilon$ and $\eta$ are small and more details in Appendix E.

## 5 Experiments

In this section, we first delve into the experimental performance of QDQ using the D4RL benchmarks [41]. Subsequently, we conduct a concise analysis of parameter settings, focusing on hyperparameter tuning across various tasks. For detailed implementation, we refer to Appendix G.

### 5.1 Performance on D4RL benchmarks for Offline RL

We evaluate the proposed QDQ algorithm on the D4RL Gym-MuJoCo and AntMaze tasks. We compare it with several strong state-of-the-art (SOTA) model-free methods: behavioral cloning (BC), BCQ [42], DT [43], AWAC [44], Onestep RL [45], TD3+BC [46], CQL [3], and IQL [10]. We also include UWAC [7], EDAC [9], and PBRL [14], which use uncertainty to pessimistically adjust the Q-value function, as well as MCQ [13], which introduces mild constraints to the Q-value function. The experimental results for the baselines reported in this paper are derived from the original experiments conducted by the authors or from replication of their official code. The reported values are normalized scores defined in D4RL [41].

Table 1: Comparison of QDQ and the other baselines on the three Gym-MuJoCo tasks. All the experiment are performed on the MuJoCo "-v2" dataset. The results are calculated over 5 random seeds.med = medium, r = replay, e = expert, ha = halfcheetah, wa = walker2d, ho=hopper

| Dataset | BC | AWAC | DT | TD3+BC | CQL | IQL | UWAC | MCQ | EDAC | PBRL | QDQ(Ours) |
|---------|------|------|-------|--------|-------|-------|------|-------|-------|-------|-------------|
| ha-med | 42.6 | 43.5 | 42.6 | 48.3 | 44.0 | 47.4 | 42.2 | 64.3 | 65.9 | 57.9 | **74.1**±1.7 |
| ho-med | 52.9 | 57.0 | 67.6 | 59.3 | 58.5 | 66.2 | 50.9 | 78.4 | **101.6** | 75.3 | **99.0**±0.3 |
| wa-med | 75.3 | 72.4 | 74.0 | 83.7 | 72.5 | 78.3 | 75.4 | 91.0 | **92.5** | 89.6 | 86.9±0.08 |
| ha-med-r | 36.6 | 40.5 | 36.6 | 44.6 | 45.5 | 44.2 | 35.9 | 56.8 | 61.3 | 45.1 | **63.7**±2.9 |
| ho-med-r | 18.1 | 37.2 | 82.7 | 60.9 | 95.0 | 94.7 | 25.3 | 101.6 | 101.0 | 100.6 | **102.4**±0.28 |
| wa-med-r | 26.0 | 27.0 | 66.6 | 81.8 | 77.2 | 73.8 | 23.6 | 91.3 | 87.1 | 77.7 | **93.2**±1.1 |
| ha-med-e | 55.2 | 42.8 | 86.8 | 90.7 | 91.6 | 86.7 | 42.7 | 87.5 | **106.3** | 92.3 | 99.3±1.7 |
| ho-med-e | 52.5 | 55.8 | 107.6 | 98.0 | 105.4 | 91.5 | 44.9 | 112.3 | 110.7 | 110.8 | **113.5**±3.5 |
| wa-med-e | 107.5 | 74.5 | 108.1 | 110.1 | 108.8 | 109.6 | 96.5 | 114.2 | 114.7 | 110.1 | **115.9**±0.2 |
| Total | 466.7 | 450.7 | 672.6 | 684.6 | 677.4 | 698.5 | 437.4 | 797.4 | 841.1 | 759.4 | **848.0**±11.8 |

Table 1 shows the performance comparison between QDQ and the baselines across Gym-MuJoCo tasks, highlighting QDQ's competitive edge in almost all tasks. Notably, QDQ excels on datasets with wide distributions, such as medium and medium-replay datasets. In these cases, QDQ effectively

avoids the problem of over-penalizing Q-values. By balancing between being too conservative and actively exploring to find the optimal Q-value function through dynamic programming, QDQ gradually converges toward the optimal Q-value, as supported by Theorem 4.4.

Table 2: Comparison of QDQ and the other baselines on the Antmaze tasks. All the experiment are performed on the Antmaze "-v0" dataset for the comparison comfortable with previous baseline. The results are calculated over 5 random seeds.

| Dataset | BC | TD3+BC | DT | Onestep RL | AWAC | CQL | IQL | QDQ(Ours) |
|---|---|---|---|---|---|---|---|---|
| umaze | 54.6 | 78.6 | 59.2 | 64.3 | 56.7 | 74.0 | 87.5 | **98.6**±2.8 |
| umaze-diverse | 45.6 | 71.4 | 53.0 | 60.7 | 49.3 | **84.0** | 62.2 | 67.8±2.5 |
| medium-play | 0.0 | 10.6 | 0.0 | 0.3 | 0.0 | 61.2 | 71.2 | **81.5**±3.6 |
| medium-diverse | 0.0 | 3.0 | 0.0 | 0.0 | 0.7 | 53.7 | 70.0 | **85.4**±4.2 |
| large-play | 0.0 | 0.2 | 0.0 | 0.0 | 0.0 | 15.8 | **39.6** | 35.6±5.4 |
| large-diverse | 0.0 | 0.0 | 0.0 | 0.0 | 1.0 | 14.9 | **47.5** | 31.2±4.5 |
| Total | 100.2 | 163.8 | 112.2 | 125.3 | 142.4 | 229.8 | 378 | **400.1**±23.0 |

Table 2 presents the performance comparison between QDQ and selected baselines[3] across AntMaze tasks, highlighting QDQ's commendable performance. While QDQ focuses on reducing overly pessimistic estimations, it does not compromise its performance on narrow datasets. This is evident in its competitive results on the medium-expert dataset in Table 1, as well as its performance on AntMaze tasks. Notably, QDQ outperforms SOTA methods on several datasets. This success is due to the inherent flexibility of the QDQ algorithm. By allowing for flexible hyperparameter control and seamless integration with various policy optimization methods, QDQ achieves a synergistic performance enhancement.

## 5.2 Parameter analysis

**The uncertainty-aware loss parameter $\alpha$.** The parameter $\alpha$ is crucial for balancing the dominance between optimistic and pessimistic updates of the Q-value (Eq.7). A higher $\alpha$ value skews updates toward the optimistic side, and we choose a higher $\alpha$ when the dataset or task is expected to be highly robust. However, the setting of $\alpha$ is also influenced by the pessimism of the Q target defined in Eq.8. For a more pessimistic Q target value, we can choose a larger $\alpha$. Interestingly, both the theoretical analyses in Theorem 4.3 and Theorem 4.4 and empirical parameter tuning suggest that variability in $\alpha$ across tasks is minimal, with a typical value around 0.95.

**The uncertainty related parameter $\beta$.** The parameter $\beta$ influences both the partitioning of high uncertainty sets and acts as a relaxation variable to control uncertainty estimation errors. When dealing with a narrow action space or a sensitive task (such as the hopper task), the value of $\beta$ should be smaller. In these cases, the Q-value is more likely to select OOD actions, increasing the risk of overestimation. This means we face greater uncertainty (Eq.8) and need to minimize overestimation errors. Therefore, we require stricter criteria to ensure actions are in-distribution and penalize the Q-values of OOD points more heavily. A detailed analysis of how to determine the value of $\beta$ can be found in Appendix G.3.

**The entropy parameter $\gamma$.** The $\gamma$ term in Eq.9 stabilizes the learning of a simple Gaussian policy, especially for action-sensitive and narrower distribution tasks. When the dataset has a wide distribution or the task shows high robustness to actions (such as in the half-cheetah task), the Q-value function generalizes better across the action space. In these cases, we can set a more lenient requirement for actions, keeping the value of $\gamma$ as small as possible or even at 0. However, when the dataset is narrow (e.g., in the AntMaze task) or when the task is sensitive to changes in actions (like in the hopper or maze tasks, where small deviations can lead to failure), a larger value of $\gamma$ is necessary. For these tasks, a simple Gaussian policy can easily sample risky actions, as it fits a single-mode policy. Nonetheless, experimental results indicate that the sensitivity of the $\gamma$ parameter is not very high. In fact, $\gamma$ in Eq.9 is relatively small compared to the Q-value, primarily to stabilize training and prevent instability in Gaussian policy action sampling. See Appendix G.7 for more details.

---

[3]We do not provide the performance of UWAC, EDAC, PBRL, and MCQ on the AntMaze task, as they do not offer useful parameter settings for this task.

# 6 Related Works

**Restrict policy deviate from OOD areas**. The distribution mismatch between the behavior policy and the learning policy can be overcome if the learning policy share the same support with the behavior policy. One approach involves explicit distribution matching constraints, where the learning policy is encouraged to align with the behavior policy by minimizing the distance between their distributions. This includes techniques based on KL-divergence [47, 48, 40, 44, 46], Jensen–Shannon divergence [49], and Wasserstein distance [47, 49]. Another line of research aims to alleviate the overly conservative nature of distribution matching constraints by incorporating distribution support constraints. These methods employ techniques such as Maximum Mean Discrepancy (MMD) distance [5], learning behavior density functions using implicit [50] or explicit [51] methods, or measuring the geometric distance between actions generated by the learning and behavior policies [52].In addition to explicit constraint methods, implicit constraints can also be implemented by learning a behavior policy sampler using techniques like Conditional Variational Autoencoders (CVAE) [42, 53, 50, 54], Autoregressive Generative Model [55], Generative Adversarial Networks (GAN) [49], normalized flow models [56], or diffusion models [57, 58, 11, 59, 11, 60].

**Pessimistic Q-value optimization**. Pessimistic Q-value methods offer a direct approach to address the issue of Q-value function overestimation, particularly when policy control fails despite the learning policy closely matching the behavior policy [3]. A promising approach to pessimistic Q-value estimation involves estimating uncertainty over the action space, as OOD actions typically exhibit high uncertainty. However, accurately quantifying uncertainty poses a challenge, especially with high-capacity function approximators like neural networks [4]. Techniques such as ensemble or bootstrap methods have been employed to estimate multiple Q-values, providing a proxy for uncertainty through Q-value variance [7, 9, 14], importance ratio [61, 62] or approximate Lower Confidence Bounds (LCB) for OOD regions [8, 9]. Other methods focus on estimating the LCB of Q-values through quantile regression [63, 34], expectile regression [10, 11], or tail risk measurement such as Conditional Value at Risk (cVAR) [12]. Alternatively, some approaches seek to pessimistically estimate Q-values based on the behavior policy, aiming to underestimate Q-values under the learning policy distribution while maximizing Q-values under the behavior policy distribution [3, 13, 64, 65]. Another category of Q-value constraint methods involves learning Q-values only within the in-sample [16, 17, 18, 19], capturing only in-sample patterns and avoid OOD risk. Furthermore, Q-value functions can be replaced by safe planning methods used in model-based RL, such as planning with diffusion models [66] or trajectory-level prediction using Transformers [43]. However, ensemble estimation of uncertainty may tend to underestimate true uncertainty, while quantile estimation methods are sensitive to Q-distribution recovery. In-sample methods may also be limited by the performance of the behavior policy.

# 7 Conclusion

We introduce QDQ, a novel framework rendering pessimistic Q-value in OOD areas by uncertainty estimation. Our approach leverages the consistency model to robustly estimate the uncertainty of Q-values. By employing this uncertainty information, QDQ can apply a judicious penalty to Q-values, mitigating the overly conservative nature encountered in previous pessimistic Q-value methods. Additionally, to enhance optimistic Q-learning within in-distribution areas, we introduce an uncertainty-aware learning objective for Q optimization. Both theoretical analyses and experimental evaluations demonstrate the effectiveness of QDQ. Several avenues for future research exist, including embedding QDQ into goal-conditioned tasks, enhancing exploration in online RL by efficient uncertainty estimation. We hope our work will inspire further advancements in offline reinforcement learning.

## Acknowledgements

We would like to thank AC and reviewers for their valuable comments on the manuscript. Bingyi Jing's research is partly supported by NSFC (No. 12371290).

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

# Appendix

## A  Further discussion about uncertainty estimation of Q-value by Q-distribution.

### A.1  The Q-value dataset enhancement with sidling window.

In Section 3.1, we analyze the challenges associated with deriving the Q-value dataset. We propose the $k$-step sliding window method to improve sample efficiency at the trajectory level. The implementation details of the $k$-step sliding window within an entire trajectory are illustrated in Figure A.1.

This sliding window framework not only facilitates the expansion of Q-value data but also prevents the state-action pairs from becoming overly dense, thereby mitigating the risk of Q-value homogenization. In continuous state-action spaces, Q-values tend to be homogeneity when state and action are close in a trajectory, which may hinder subsequent learning of the Q-value distribution.

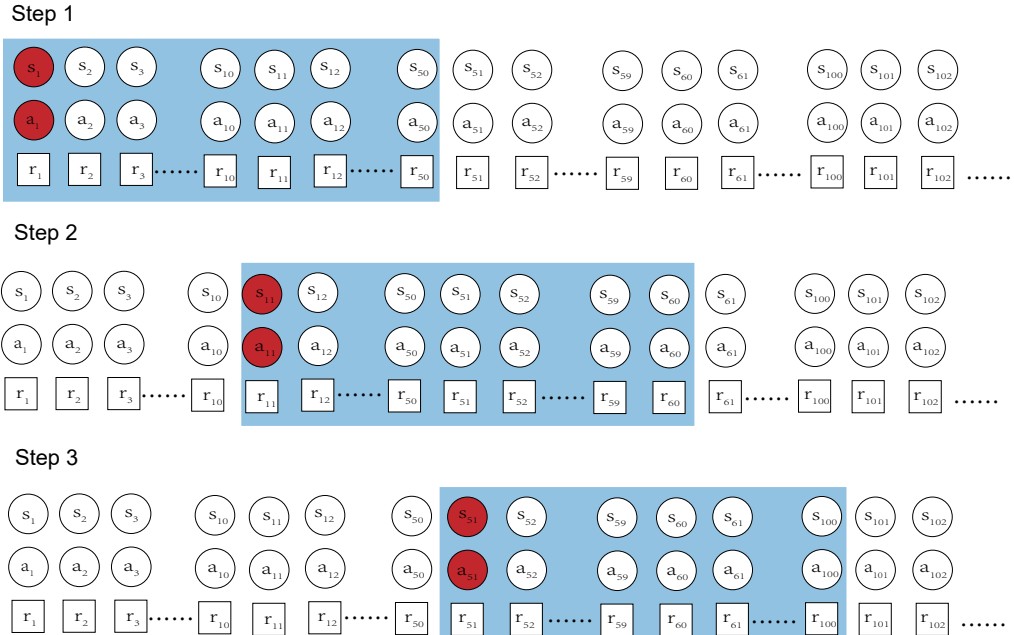

Figure A.1: This exemplifies how the sliding window mechanism operates to augment Q data. Let's consider a sliding window with a width of 50 and a step size of $k = 10$. For a specific trajectory, at step 1, we commence with $(s_1, a_1)$ and compute the truncated Q-value utilizing trajectories within the window. At step 2, the sliding window progresses $k$ steps forward, allowing us to compute the truncated Q-value for $(s_{1+k}, a_{1+k})$.

### A.2  Drawbacks of the diffusion model for estimating the uncertainty

Suppose we use the score matching method proposed in [30] to learn a conditional score network $s_\theta(x, \sigma|s, a)$ to approximate the score function of the Q-value distribution $p_Q(x)$. Then we use the annealed Langevin dynamics as in [30] to sample from the learned Q-value distribution. For $x_0 \sim \pi(x)$ from some arbitrary prior distribution $\pi(x)$, the denoised sample is:

$$x_{i+1} = x_i + \epsilon \cdot s_\theta(x_i, \sigma|s, a) + \sqrt{2\epsilon}z_i, i = 0, 1, 2, \cdots, T. \quad z_i \sim \mathcal{N}(0, 1). \quad (A.1)$$

The distribution of $x_{i+1}$ equals $p_Q(x)$ when $\epsilon \to 0$ and $T \to \infty$.

Our primary approach to quantify the uncertainty of the action with respect to the Q-value is to evaluate the spread of the sampled Q-values for each $(s, a)$. We then use the gradient of the Q-value samples with respect to the action to assess their sensitivity to changes in action. By iteratively deriving the gradient over the sampling chain of length $T$, as shown in Eq.A.1, we approximate the following outcome:

$$\frac{\partial x_T}{\partial a} = \sum_{i=1}^{T} c_i \epsilon^i \frac{\partial^i s_\theta(\cdot|s, a)}{\partial a^i}. \tag{A.2}$$

From Eq.A.2, the impact of actions on Q-value samples learned from the diffusion model shows considerable instability, especially during the iterative gradient-solving process for the score network $s_\theta(x, \sigma|s, a)$. This instability often leads to gradient vanishing or exploding. While the diffusion model effectively recovers the Q-value distribution with high precision, the multi-step sampling process introduces significant fluctuations and noise, making it difficult to accurately assess the uncertainty of Q-values for different actions.

Additionally, as different prior information may yield the same target sample, and this stochastic correlation also introduces an uncontrollable impact on uncertainty of the Q-value. However, the effect of prior sample variance on the uncertainty of the sampled Q-value can not be quantified with the diffusion model. So we can not guarantee if the absolute influence on the Q-value uncertainty is from the action, then the performance of the uncertainty can not be guaranteed.

## B Convergence of the truncated Q-value distribution.

We first give a formal introduction of Theorem 4.1 as Theorem B.1.

**Theorem B.1.** *Suppose the true distribution of Q-value w.r.t the behavior policy $\pi_\beta$ is defined as $F_{Q^{\pi_\beta}}(x)$. By Eq.5, we derive the truncated Q-value $Q_{\mathcal{T}}^{\pi_\beta}$, denote the distribution of the truncated Q-value is $F_{Q_{\mathcal{T}}^{\pi_\beta}}(x)$. Assume the true Q-value is finite over the state and action space,then $Q_{\mathcal{T}}^{\pi_\beta}$ converge in-distribution to the true true Q-value $Q^{\pi_\beta}$.*

$$F_{Q_{\mathcal{T}}^{\pi_\beta}}(x) \to F_{Q^{\pi_\beta}}(x), \mathcal{T} \to +\infty. \tag{B.1}$$

We will show the distribution of the truncated Q-value $Q_{\mathcal{T}}^{\pi_\beta}$ has the same property with the true distribution of the Q-value $Q_{\mathcal{T}}^{\pi_\beta}$ and give a brief proof of Theorem 4.1.

Suppose the state-action space determined by the offline RL dataset $\mathcal{D}$ is $\Omega_{\mathcal{D}} = \mathcal{S}_{\mathcal{D}} \times \mathcal{A}_{\mathcal{D}}$. Note that $Q^{\pi_\beta}$ can be seen as an r.v. defined as: $(\Omega_{\mathcal{D}}, \mathscr{F}, \mathbb{P} \cdot \pi_\beta) \xrightarrow{Q^{\pi_\beta}} (\mathbb{R}, \mathcal{B}(\mathbb{R}), (\mathbb{P} \cdot \pi_\beta) \circ (Q^{\pi_\beta})^{-1})$, where $\mathscr{F}$ is the $\sigma$-field on $\Omega_{\mathcal{D}}$, $\mathbb{P} \cdot \pi_\beta$ is the probability measure on $\Omega_{\mathcal{D}}$ and $\mathbb{P}$ is the transition probability measure over state, $\mathcal{B}(\mathbb{R})$ is the Borel $\sigma$-field on $\mathbb{R}$, $(\mathbb{P} \cdot \pi_\beta) \circ (Q^{\pi_\beta})^{-1} = (\mathbb{P} \cdot \pi_\beta)((Q^{\pi_\beta})^{-1})$ is the push forward probability measure on $\mathbb{R}$. Same as $Q^{\pi_\beta}$, $Q_{\mathcal{T}}^{\pi_\beta}$ can also be seen as a r.v..

Then we show that $Q_{\mathcal{T}}^{\pi_\beta}$ converge to $Q^{\pi_\beta}$ almost surely: $Q_{\mathcal{T}}^{\pi_\beta} \xrightarrow{a.s.} Q^{\pi_\beta}$, when sending $\mathcal{T}$ to infinity.

Define the trajectory level dataset $\mathcal{D}_\tau = \{\tau_k | \tau_k = (s_{k_0}, a_{k_0}, r_{k_0}, s_{k_1}, a_{k_1}, r_{k_1}, \cdots)\}$. Then for any trajectory $\tau_k \in \mathcal{D}_\tau$, the true Q-value w.r.t this trajectory can be rewrite without the expectation as:$Q^\pi(s_{k_0}, a_{k_0}) = \sum_{j=1}^{\infty} \gamma^{j-1} r(s_{k_j}, a_{k_j})$.

**Truncating Q-value by termination situation.** If the terminal occurs at step $k_t$ of the trajectory $\tau_k$, then we have $r(s_{k_j}, a_{k_j}) = 0$, for $j > k_t$. Then the truncated Q-value is identical the true Q-value: $Q_{k_t}^{\pi_\beta} \equiv Q^\pi(s_{k_0}, a_{k_0})$. So the distribution of these two distribution is same for the situation when the truncation is happened due to terminal of the task.

**Truncating Q-value by sliding window situation.** If the Q-value is truncated by a sliding window as shown in Figure A.1, then for a specific $k$- step sliding widow of width $\mathcal{T}$ with starting point $(s_i, a_i)$ over a trajectory $\tau$, we have $Q_{\mathcal{T}}^{\pi_\beta}(s_i, a_i) = \sum_{m=i}^{\mathcal{T}} \gamma^{m-1} r(s_m, a_m)$.

Define the state action set $B_n(\xi) := \overset{\infty}{\underset{\mathcal{T}=n}{\cup}} A_n(\xi)$, where $A_{\mathcal{T}}(\xi) = \{(s,a) : |Q_{\mathcal{T}}^{\pi_\beta}(s,a) - Q^{\pi_\beta}(s,a)| > \xi\}$. By the definition of Q-value, we have $B_m(\xi) = A_m(\xi)$, as $A_{\mathcal{T}}(\xi) \supset A_{\mathcal{T}+1}(\xi) \supset A_{\mathcal{T}+2}(\xi) \supset \cdots$ is decreasing. So $B_n(\xi)$ is also decreasing. Then,

$$\lim_{m \to \infty} \mathbb{P}(B_n(\xi)) = \mathbb{P}(\lim_{n \to \infty} B_n(\xi)) = \mathbb{P}(\lim_{n \to \infty} A_n(\xi)) \tag{B.2}$$

By definition, $A_n(\xi) = \{(s,a) : |Q_n^{\pi_\beta}(s,a) - Q^{\pi_\beta}(s,a)| > \xi\}$, and assume the reward function $r(\cdot, \cdot)$ is bounded as $r(\cdot, \cdot) < c$ for some constant $c$,

$$|Q_n^{\pi_\beta}(s,a) - Q^{\pi_\beta}(s,a)| = \sum_{k=n+1}^{\infty} \gamma^n r(s_k, a_k) \leq c \sum_{k=n+1}^{\infty} \gamma^n = \frac{\gamma^n}{1-\gamma} \cdot c \tag{B.3}$$

So $A_n(\xi) \to \emptyset$ when sending $n$ to infinity, as the discount factor $\gamma < 1$ by definition.

Then by Eq.B.2, we have $\lim_{m \to \infty} \mathbb{P}(B_n(\xi)) = 0, \forall \xi > 0$. In probability theory, this is equivalent to $Q_{\mathcal{T}}^{\pi_\beta} \xrightarrow{a.s.} Q^{\pi_\beta}$.

Furthermore, if $Q_{\mathcal{T}}^{\pi_\beta}$ converge to $Q^{\pi_\beta}$ almost surely, then $Q_{\mathcal{T}}^{\pi_\beta}$ also converge to $Q^{\pi_\beta}$ in probability, and in-distribution. Hence, we finish the proof.

*Remark* B.1. Theorem 4.1 suggests that for arbitrary small $\epsilon$, there exists a sufficiently large $\mathcal{T}$, such that $|F_{Q_{\mathcal{T}}^{\pi_\beta}}(x) - F_{Q^{\pi_\beta}}(x)| < \epsilon$. Given that the impact of rewards diminishes exponentially after as the increasing of trajectory length, it is unnecessary to set $\mathcal{T}$ to an excessively large value. It's important to remember that the goal of the Q dataset is to learn the Q-distribution and assess the uncertainty of different actions. Therefore, the absolute magnitude of Q is not crucial. Additionally, using too many future steps may introduce significant uncertainty into the Q-value, as predictions for the distant future can be inaccurate.

*Remark* B.2. During the proof, the specific starting point of the sliding window holds no significance; rather, our focus lies solely on the length of the window. This is primarily due to the Markovian nature of trajectories in RL, where the current state and action are unaffected by previous ones and adhere to a memoryless property. Consequently, the starting point of the sliding window exerts minimal influence on the computation of Q.

## C    Robustness of consistency model for uncertainty measure.

The formal introduction of Theorem 4.2 is shown in Theorem C.1.

**Theorem C.1.** *Follow the assumptions in [20], we assume $f_\theta(x, T|(s,a))$ is L-Lipschitz.*

*By using the partial gradient to analyze the influence of prior samples $\hat{x}_T$, time step $T$ and action $a$ to the variance of the denoised sample $var(X_\epsilon)$, with high probability, we have:*

(1) *Prior noise influence the variance $V(X_\epsilon)$ is bounded by:* $|\frac{\partial V(X_\epsilon)}{\partial \hat{x}_T}| = O(L^2 T n^{-1}\sqrt{T \log(n)})\mathbf{1}$.

(2) *Time step $T$ influence the variance $V(X_\epsilon)$ is bounded by:* $|\frac{\partial var(X_\epsilon)}{\partial T}| = O(L^2 T \sqrt{\log n})$.

(3) *Action $a$ influence the variance $V(X_\epsilon)$ is bounded by:* $|\frac{\partial var(X_\epsilon)}{\partial a}| = O(L^2 T \sqrt{\log n})\mathbf{1}$.

As discussed in Section 3.1 and A.2, while the diffusion model has shown great success in learning distributions and generating samples, it is less suitable for scenarios where the influence of certain parameters on sample uncertainty must be guaranteed. When estimating uncertainty, we sample multiple Q-values for each $(s,a)$ pair, and the standard deviation of these sampled Q-values measures the uncertainty. Therefore, it is crucial that the sample spread is sensitive to changes in action to accurately judge OOD actions.

However, the multi-step forward denoising process of the diffusion model undermines the influence of actions on the sampled Q-values, compromising the robustness of uncertainty estimation. Additionally, the lack of a one-to-one correspondence between prior information and target samples prevents the cancellation of prior effects on the Q-value distribution through repeated sampling.

In contrast, the consistency model addresses these challenges. It not only overcomes the aforementioned issues, but its one-step sampling significantly enhances efficiency. In the following theoretical analysis, we will demonstrate the robustness of the consistency model in estimating uncertainty.

As described in [20] and Section 2.2, a consistency model $f_\theta(x, t)$ is trained to mapping the prior noise on any trajectory of PF ODE to the trajectory's origin $x_\epsilon$ by: $f_\theta(x, t) = x_\epsilon$, given $x$ and $x_\epsilon$ belong to the same PF ODE trajectory. $f_\theta(x, t)$ is defined as in Eq.4.

Suppose we trained a conditional consistency model $f_\theta(x, t|s, a)$ with the truncated Q-value dataset $D_Q = \{Q_\mathcal{T}^{\pi_\beta}(s, a)\}$, following the one-step sampling of consistency model, we first initial $n$ noise $\hat{x}_{T_i} \sim \mathcal{N}(0, T^2), i = 1, 2, ..., n$, then do one-step forward denosing and derive $n$ sample $\hat{x}_{\epsilon_i} = f_\theta(\hat{x}_{T_i}, T|s, a)$, where $T$ is a fixed time step. The variance based on the Q sample is:

$$V(X_\epsilon) = \frac{1}{n-1} \sum_{i=1}^{n} \left[ f_\theta(\hat{x}_{T_i}, T|(s, a)) - \frac{1}{n} \sum_{i=1}^{n} f_\theta(\hat{x}_{T_i}, T|(s, a)) \right]^2. \tag{C.1}$$

Next, we derive the gradient of $V(X_\epsilon)$ w.r.t $\hat{x}_{T_i}$, $T$, $a$, and check how change in these variable influence the variance. As state $s$ is always in-distribution during offline RL training process and has little influence on the uncertainty of the sampled Q-value, we skip the analysis.

Following [20], we assume that $f_\theta(x, t|s, a)$ is $L$-Lipschitz bounded, i.e., for any $x$ and $y$,

$$\|f_\theta(x, t|(s, a)) - f_\theta(y, t|(s, a))\|_2 \leq L \|x - y\|_2.$$

**Gradient of the prior $\hat{x}_{T_i}$.**

*Proof.* Let $e_i$ be the square difference of the $i$-th prior $\hat{x}_{T_i}$:

$$e_i(\hat{x}_{T_i}) := \left[ f_\theta(\hat{x}_{T_i}, T|(s, a)) - \frac{1}{n} \sum_{i=1}^{n} f_\theta(\hat{x}_{T_i}, T|(s, a)) \right]^2. \tag{C.2}$$

Then we have:

$$\frac{\partial V(X_\epsilon)}{\partial \hat{x}_{T_i}} = \frac{1}{n-1} \sum_{j=1}^{n} \frac{\partial e_j(\hat{x}_{T_j})}{\partial \hat{x}_{T_i}} \tag{C.3}$$

If $j = i$,

$$\frac{\partial e_j(\hat{x}_{T_j})}{\partial \hat{x}_{T_i}} = 2 \left[ f_\theta(\hat{x}_{T_i}, T|(s, a)) - \frac{1}{n} \sum_{i=1}^{n} f_\theta(\hat{x}_{T_i}, T|(s, a)) \right] \cdot \left[ \frac{\partial f_\theta(\hat{x}_{T_i}, T|(s, a))}{\partial \hat{x}_{T_i}} - \frac{1}{n} \frac{\partial f_\theta(\hat{x}_{T_i}, T|(s, a))}{\partial \hat{x}_{T_i}} \right]$$

$$= 2 \left[ f_\theta(\hat{x}_{T_i}, T|(s, a)) - \frac{1}{n} \sum_{i=1}^{n} f_\theta(\hat{x}_{T_i}, T|(s, a)) \right] \cdot \frac{n-1}{n} \frac{\partial f_\theta(\hat{x}_{T_i}, T|(s, a))}{\partial \hat{x}_{T_i}}. \tag{C.4}$$

If $j \neq i$,

$$\frac{\partial e_j(\hat{x}_{T_j})}{\partial \hat{x}_{T_i}} = 2 \left[ f_\theta(\hat{x}_{T_j}, T|(s, a)) - \frac{1}{n} \sum_{i=1}^{n} f_\theta(\hat{x}_{T_i}, T|(s, a)) \right] \cdot \left[ 0 - \frac{1}{n} \frac{\partial f_\theta(\hat{x}_{T_i}, T|(s, a))}{\partial \hat{x}_{T_i}} \right]$$

$$= 2 \left[ f_\theta(\hat{x}_{T_j}, T|(s, a)) - \frac{1}{n} \sum_{i=1}^{n} f_\theta(\hat{x}_{T_i}, T|(s, a)) \right] \cdot -\frac{1}{n} \frac{\partial f_\theta(\hat{x}_{T_i}, T|(s, a))}{\partial \hat{x}_{T_i}}. \tag{C.5}$$

Thus, plugging Eq.C.4 and Eq.C.5 into $\frac{\partial V(X_\epsilon)}{\partial \hat{x}_{T_i}}$ yields

$$\frac{\partial V(X_\epsilon)}{\partial \hat{x}_{T_i}} = \frac{1}{n-1} \sum_{j=1}^{n} \frac{\partial e_j(\hat{x}_{T_j})}{\partial \hat{x}_{T_i}}$$

$$= \frac{2}{n(n-1)} \frac{\partial f_\theta(\hat{x}_{T_i}, T|(s, a))}{\partial \hat{x}_{T_i}} \left[ (n-1) f_\theta(\hat{x}_{T_i}, T|(s, a)) - \sum_{j \neq i} f_\theta(\hat{x}_{T_j}, T|(s, a)) \right]. \tag{C.6}$$

As $f_\theta(x, t|s, a)$ is $L$-Lipschitz bounded, we have

$$|\frac{\partial V(X_\epsilon)}{\partial \hat{x}_{T_i}}| \leq \frac{2}{n(n-1)}|\frac{\partial f_\theta(\hat{x}_{T_i}, T|(s, a))}{\partial \hat{x}_{T_i}}|\sum_{j \neq i}|f_\theta(\hat{x}_{T_i}, T|(s, a)) - f_\theta(\hat{x}_{T_j}, T|(s, a))|$$

$$\leq \frac{2}{n(n-1)} \cdot L \cdot L \sum_{j \neq i}|\hat{x}_{T_i} - \hat{x}_{T_j}| \leq \frac{2}{n} \cdot L^2 \cdot c\sqrt{\log n} \cdot T, \quad (C.7)$$

where $|\hat{x}_{T_i} - \hat{x}_{T_j}|, \forall j \neq i$ can be bounded by $cT\sqrt{\log n}$ due to $\hat{x}_{T_i} \sim \mathcal{N}(0, T^2)$ with probability at least $1 - n^{-1}$. Denote the constant with $c_p$ and apply the previous process to all the prior samples completes the proof.

**Gradient of the time step $T$.**

*Proof.*

Note that

$$\frac{\partial V(X_\epsilon)}{\partial T} = \frac{1}{n-1}\sum_{j=1}^{n}\frac{\partial e_j(T)}{\partial T}. \quad (C.8)$$

Taking partial gradient of $e_i(T)$ w.r.t $T$ for any $j \in \{1, 2, \cdots, n\}$, we obtain

$$\left|\frac{\partial e_j(\hat{x}_{T_j})}{\partial T}\right|$$

$$= 2\left|\left[f_\theta(\hat{x}_{T_j}, T|(s, a)) - \frac{1}{n}\sum_{i=1}^{n}f_\theta(\hat{x}_{T_i}, T|(s, a))\right] \cdot \left[\frac{\partial f_\theta(\hat{x}_{T_j}, T|(s, a))}{\partial T} - \frac{1}{n}\sum_{i=1}^{n}\frac{\partial f_\theta(\hat{x}_{T_i}, T|(s, a))}{\partial T}\right]\right|$$

$$\leq 4\frac{L^2}{n}\sum_{i=1}^{n}|\hat{x}_{T_j} - \hat{x}_{T_i}| \leq 4L^2\sqrt{\log n}, \quad (C.9)$$

with probability at least $1 - n^{-1}$, since $|\hat{x}_{T_i} - \hat{x}_{T_j}|, \forall j \neq i$ can be bounded by $cT\sqrt{\log n}$ due to $\hat{x}_{T_i} \sim \mathcal{N}(0, T^2)$ with probability at least $1 - n^{-1}$.

Plugging Eq.C.9 into Eq.C.8 yields

$$\left|\frac{\partial V(X_\epsilon)}{\partial T}\right| \leq 4cL^2T\sqrt{\log n}. \quad (C.10)$$

**Gradient of the action $a$.**

For the gradient of $\frac{\partial V(X_\epsilon)}{\partial a}$, we just need to take partial gradient of $V(X_\epsilon)$ for each dimmension of the action $a = \{a_1, a_2, \cdots, a_m\}$ by $\frac{\partial V(X_\epsilon)}{\partial a_i}$. The result is same as those we got in Eq.C.10.

Then take the vector form, we have $|\frac{\partial V(X_\epsilon)}{\partial a}| = O(L^2T\sqrt{\log n}) \cdot \mathbf{1}$, which finish the proof.

*Remark* C.1. Theorem 4.2 elucidates the diminishing impact of random prior on the variance of the denoised Q-value as the sample size increases. Leveraging the consistency of sampling, we mitigate concerns regarding the influence of a priori samples on the uncertainty of final target samples. Given the fixed sampling step size $T$, we also address concerns about its effect on the uncertainty of Q samples. However, for a thorough analysis, we still include the gradient analysis of $V(X_\epsilon)$ against $T$.

*Remark* C.2. The influence of actions on sample variance depends on factors like the Lipschitz factor and sample size. As the sample size increases, the impact of actions on Q sample variance does not diminish; instead, it becomes more sensitive. Larger Q sample variance is more likely to occur when OOD actions are present. Consequently, the consistency model proves to be a reliable approach for Q-value sampling and uncertainty estimation.

*Remark* C.3. Although experiments in [20] show that the performance of the consistency model is less competitive compared to the diffusion model or adversarial generators like GANs, these findings have minimal relevance to our method. Our primary focus is not on the absolute accuracy of the sampled Q-values but on the sensitivity of Q sample dispersion to OOD actions and its ability to effectively capture uncertainty in such cases. Additionally, the high sampling efficiency achieved through one-step sampling compensates for the minor performance discrepancies of the consistency model.

# D  Convergence of the uncertainty-aware learning objective for recovering the Q-value function.

We first give a formal introduction of Theorem 4.3 below.

**Theorem D.1.** *Updating Q-value $Q_\theta(s, a)$ via the uncertainty-aware objective in Eq.7 is equivalent to minimizing the $L_2$-norm of Bellman residuals:* $\mathbb{E}_{s \sim P_\mathcal{D}(s), a \sim \pi(a|s)}[Q(s, a) - \mathscr{F}Q(s, a)]^2$, *where the Bellman operator $\mathscr{F}Q(s, a)$ is defined as:*

$$\mathscr{F}Q(s, a) := r(s, a) + \gamma \mathbb{E}_{s' \sim P_\mathcal{D}(s')}\{\max_{a'}[\alpha Q(s', a') + (1 - \alpha)Q_L(s', a')]\}. \tag{D.1}$$

*In addition, assume $\frac{1}{\mathcal{H}_Q(a'|s')}1_{(a' \in \mathcal{U}(Q))} < \beta$. Then the Bellman operator $\mathscr{F}Q$ is $c\gamma$-contraction operator in the $L_\infty$ norm, where $c < 1$. The Q-value function $Q_\theta(s, a)$ can converge to a fixed point by value iteration method.*

We will show the the uncertainty-aware learning objective is equivalent to minimized the Bellman equation defined by a specific Bellman operator $\mathscr{F}Q$ firstly.

Then the we proof the Bellman operator $\mathscr{F}Q$ is $c\gamma$-contraction operator in the $L_\infty$ norm, where $c < 1$. The Q-value function $Q_\theta(s, a)$ can converge to a fixed point by the value iteration method.

## D.1  Derivation of the Bellman operator $\mathscr{F}Q$.

Recall the Q-value is optimized by the uncertainty-aware learning objective by Eq.7:

$$\mathcal{L}_{uw}(Q_\theta) = \min_\theta \{\alpha \mathcal{L}(Q_\theta)_H + (1 - \alpha)\mathcal{L}(Q_\theta)_L\}, \tag{D.2}$$

where

$$\begin{aligned}
\mathcal{L}(Q)_H :=& \mathbb{E}_{s \sim P_\mathcal{D}(s), a \sim \pi(a|s)}[Q_\theta(s, a) - (\mathcal{B}Q)(s, a)]^2 \\
=& \mathbb{E}_{s \sim P_\mathcal{D}(s), a' \sim \pi(a|s)}[Q_\theta(s, a) - (r(s, a) + \gamma \mathbb{E}_{s' \sim P_\mathcal{D}(s')}[\max_{a'} Q_\theta(s', a')])]^2, \tag{D.3}
\end{aligned}$$

$$\begin{aligned}
\mathcal{L}(Q)_L :=& \mathbb{E}_{s \sim P_\mathcal{D}(s), a \sim \pi(a|s)}[Q_\theta(s, a) - (\mathcal{B}Q)_L(s, a)]^2 \\
=& \mathbb{E}_{s \sim P_\mathcal{D}(s), a' \sim \pi(a|s)}[Q_\theta(s, a) - (r(s, a) + \gamma \mathbb{E}_{s' \sim P_\mathcal{D}(s')}[\max_{a'} Q_L(s', a')])]^2. \tag{D.4}
\end{aligned}$$

The uncertainty penalized Q target $Q_L(s', a')$ is defined as:

$$Q_L(s', a') = \frac{1}{\mathcal{H}_Q(a'|s')}Q_\theta(s', a')1_{(a' \in \mathcal{U}(Q))} + \beta Q_\theta(s', a')1_{(a' \notin \mathcal{U}(Q))}. \tag{D.5}$$

For simplicity, we ignore the estimation parameter of $Q_\theta(s, a)$ and just use $Q(s, a)$ in the following proof.

We can just take the uncertainty-aware loss in Eq.7 as a plain regression like loss and we have:

$$\begin{aligned}
\alpha \mathcal{L}(Q)_H + (1 - \alpha)\mathcal{L}(Q)_L =& \alpha[Q(s, a) - (\mathcal{B}Q)(s, a)]^2 + (1 - \alpha)[Q(s, a) - (\mathcal{B}Q)_L(s, a)]^2 \\
=& \alpha[Q(s, a)^2 - 2Q(s, a)(\mathcal{B}Q)(s, a) + ((\mathcal{B}Q)(s, a))^2] \\
&+ (1 - \alpha)[Q(s, a)^2 - 2Q(s, a)(\mathcal{B}Q)_L(s, a) + ((\mathcal{B}Q)_L(s, a))^2] \\
=& \alpha Q(s, a)^2 - \alpha 2Q(s, a)(\mathcal{B}Q)(s, a) + \alpha((\mathcal{B}Q)(s, a))^2 \\
&+ (1 - \alpha)Q(s, a)^2 - (1 - \alpha)2Q(s, a)(\mathcal{B}Q)_L(s, a) + (1 - \alpha)((\mathcal{B}Q)_L(s, a))^2 \\
=& Q(s, a)^2 - 2Q(s, a)[\alpha(\mathcal{B}Q)(s, a) + (1 - \alpha)(\mathcal{B}Q)_L(s, a)] \\
&+ \alpha((\mathcal{B}Q)(s, a))^2 + (1 - \alpha)((\mathcal{B}Q)_L(s, a))^2 \\
=& [Q(s, a) - (\alpha(\mathcal{B}Q)(s, a) + (1 - \alpha)(\mathcal{B}Q)_L(s, a))]^2 + C, \tag{D.6}
\end{aligned}$$

where $C$ is a factor that not related to $Q(s, a)$, since the value of $(\mathcal{B}Q)_L(s, a)$ and $(\mathcal{B}Q)(s, a)$ are fixed as we update $Q$. By the definition of $(\mathcal{B}Q)(s, a)$ and $(\mathcal{B}Q)_L(s, a)$ in Eq.D.3 and Eq.D.4, the uncertainty-aware learning is equivalent to minimized the following Bellman equation:

$$\mathbb{E}_{s \sim P_\mathcal{D}(s), a \sim \pi(a|s)}[Q(s, a) - (\mathscr{F}Q)(s, a)]^2, \tag{D.7}$$

where the specific Bellman operator $(\mathscr{F}Q)(s, a)$ is defined in Eq.D.1. Then we finish the proof.

## D.2 Bellman operator $\mathscr{F}Q$ is $c\gamma$-contraction operator in the $L_\infty$ norm.

Then we give a brief proof of the convergence of the Bellman operator $\mathscr{F}Q$ by value interative optimization.

For any Q-value function $Q(s,a)$, $Q'(s,a)$, define $I = |\mathscr{F}Q(s,a) - \mathscr{F}Q'(s,a)|$, we have

$$
\begin{aligned}
I =& |r(s,a) + \gamma \mathbb{E}_{s' \sim P_\mathcal{D}(s')}\{\max_{a'}[\alpha Q(s',a') + (1-\alpha)Q_L(s',a')]\} - \\
&+ (r(s,a) + \gamma \mathbb{E}_{s' \sim P_\mathcal{D}(s')}\{\max_{a'}[\alpha Q'(s',a') + (1-\alpha)Q'_L(s',a')]\})| \\
=& \gamma |\mathbb{E}_{s' \sim P_\mathcal{D}(s')} \max_{a'}\{\alpha Q(s',a') + (1-\alpha)Q_L(s',a') - [\alpha Q'(s',a') + (1-\alpha)Q'_L(s',a')]\}| \\
\leq& \gamma \mathbb{E}_{s' \sim P_\mathcal{D}(s')} |\max_{a'}\{\alpha Q(s',a') + (1-\alpha)Q_L(s',a') - [\alpha Q'(s',a') + (1-\alpha)Q'_L(s',a')]\}| \\
\leq& \gamma \max_{s'} |\max_{a'}\{\alpha Q(s',a') + (1-\alpha)Q_L(s',a') - [\alpha Q'(s',a') + (1-\alpha)Q'_L(s',a')]\}| \\
\leq& \gamma \max_{s'} \max_{a'} |\alpha Q(s',a') + (1-\alpha)Q_L(s',a') - [\alpha Q'(s',a') + (1-\alpha)Q'_L(s',a')]| \\
=& \gamma \max_{s'} \max_{a'} |\alpha(Q(s',a') - Q'(s',a')) + (1-\alpha)(Q_L(s',a') - Q'_L(s',a'))| \\
=& \gamma \max_{s'} \max_{a'} |\alpha(Q(s',a') - Q'(s',a')) + (1-\alpha)(\beta Q(s',a') - \beta Q'(s',a'))1_{(a' \notin \mathcal{U}(Q))} \\
&+ (1-\alpha)(\frac{1}{\mathcal{H}_Q(a'|s')}Q(s',a') - \frac{1}{\mathcal{H}_Q(a'|s')}Q'(s',a'))1_{(a' \in \mathcal{U}(Q))}| \\
=& \gamma \max_{s'} \max_{a'} |\alpha(Q(s',a') - Q'(s',a')) + (1-\alpha)\beta(Q(s',a') - Q'(s',a'))1_{(a' \notin \mathcal{U}(Q))} \\
&+ \frac{(1-\alpha)}{\mathcal{H}_Q(a'|s')}(Q(s',a') - Q'(s',a'))1_{(a' \in \mathcal{U}(Q))}| \\
=& \gamma \max_{s'} \max_{a'} |\alpha(Q(s',a') - Q'(s',a'))1_{(a' \notin \mathcal{U}(Q))} + (1-\alpha)\beta(Q(s',a') - Q'(s',a'))1_{(a' \notin \mathcal{U}(Q))} \\
&+ \alpha(Q(s',a') - Q'(s',a'))1_{(a' \in \mathcal{U}(Q))} + \frac{(1-\alpha)}{\mathcal{H}_Q(a'|s')}(Q(s',a') - Q'(s',a'))1_{(a' \in \mathcal{U}(Q))}| \\
\leq& \gamma \max_{s'} \max_{a'}\{(\alpha + (1-\alpha)\beta)|Q(s',a') - Q'(s',a')|1_{(a' \notin \mathcal{U}(Q))} + (\alpha + \frac{(1-\alpha)}{\mathcal{H}_Q(a'|s')})|Q(s',a') - Q'(s',a')|1_{(a' \in \mathcal{U}(Q))}\} \\
\leq& \gamma \max_{s'} \max_{a'}\{\max\{\alpha + (1-\alpha)\beta, \alpha + \frac{(1-\alpha)}{\mathcal{H}_Q(a'|s')}\}|Q(s',a') - Q'(s',a')|\} \\
=& \gamma \max\{\alpha + (1-\alpha)\beta, \alpha + \frac{(1-\alpha)}{\mathcal{H}_Q(a'|s')}1_{(a' \in \mathcal{U}(Q))}\}||Q(s',a') - Q'(s',a')||_\infty \quad\quad\quad \text{(D.8)}
\end{aligned}
$$

Since $\frac{1}{\mathcal{H}_Q(a'|s')}1_{(a' \in \mathcal{U}(Q))} < \beta$, we have $\max\{\alpha + (1-\alpha)\beta, \alpha + \frac{(1-\alpha)}{\mathcal{H}_Q(a'|s')}\} = \alpha + (1-\alpha)\beta$ is always true. As $\beta < 1$, then $\alpha + (1-\alpha)\beta < \alpha + (1-\alpha) < 1$.

Set $c = \alpha + (1-\alpha)\beta$, then we have

$$|\mathscr{F}Q(s,a) - \mathscr{F}Q'(s,a)| \leq c\gamma ||Q(s',a') - Q'(s',a')||_\infty, \quad\quad\quad \text{(D.9)}$$

which implies $\mathscr{F}Q$ is $c\gamma$-contraction operator with $c\gamma < 1$.

Suppose $Q^\Delta$ is the stationary point of the Bellman equation in Eq.D.7, then it can be shown that $Q$ iteratively updated by Eq.D.7 can converge to $Q^\Delta$:

$$
\begin{aligned}
||Q^{t+1} - Q^\Delta||_\infty =& ||\mathscr{F}Q^t - \mathscr{F}Q^\Delta||_\infty \leq c\gamma ||Q^t - Q^\Delta||_\infty \\
\leq& (c\gamma)^2 ||Q^{t-1} - Q^\Delta||_\infty \\
\leq& \cdots \leq (c\gamma)^{t+1} ||Q^0 - Q^\Delta||_\infty. \quad\quad\quad \text{(D.10)}
\end{aligned}
$$

Sending $t$ to infinity, we can derive $Q^t$ converge to $Q^\Delta$, then we finish the proof.

*Remark* D.1. The assumption $\frac{1}{\mathcal{H}_Q(a'|s')}1_{(a' \in \mathcal{U}(Q))} < \beta$ can be always satisfied. Roughly speaking, since $a' \in \mathcal{U}(Q)$, the uncertainty of Q-value $\mathcal{H}_Q(a'|s')$ on this $a'$ has large uncertainty due to the OOD property. Furthermore, we can scale the absolute value $\mathcal{H}_Q(a'|s')$ for all the action with

same factor to guarantee $\frac{1}{\mathcal{H}_Q(a'|s')}1_{(a'\in\mathcal{U}(Q))} < \beta$ without hurting the relative comparison for the uncertainty. Furthermore, experiment results have shown that $\frac{1}{\mathcal{H}_Q(a'|s')}1_{(a'\in\mathcal{U}(Q))} < \beta$ is consistently satisfied without additional processing.

# E    Performance of the Q-value function $Q^k(s,a)$ derived by QDQ.

In this section, we delve into an analysis of the performance of the Q-value function derived from the QDQ algorithm. Given that the primary aim of QDQ is to mitigate the issue of excessively conservative in most pessimistic Q-value methods, our focus is directed towards scrutinizing the disparity between the optimal Q-value within the offline RL framework and the optimal Q-value function yielded by QDQ.

The following is a formal version of Theorem 4.4.

**Theorem E.1.** *Suppose the optimal Q-value function over state-action space $\mathcal{S}_\mathcal{D} \times \mathcal{A}_\mathcal{D}$ defined by the dataset $\mathcal{D}$ is $Q^*$. Then with probability at least $1 - \eta$, the Q-value function $Q^\Delta$ learned by minimizing uncertainty-aware loss (Eq.7) can approach the optimal $Q^*$ with a small constant:*

$$\left\|Q^\Delta - Q^*\right\|_\infty \le \epsilon, \tag{E.1}$$

*where $\epsilon$ is error rate related to the difference between the classical Bellman operator $\mathcal{B}Q$ and the QDQ Bellman operator $\mathscr{F}Q$, and $\eta$ is determined by the probability that $\max_{a'}\{(1 - \beta))|Q(s',a')|1_{(a'\notin\mathcal{U}(Q))} + (1 - \frac{1}{\mathcal{H}_Q(a'|s')})|Q(s',a')|1_{(a'\in\mathcal{U}(Q))}\} = \max_{a'}\{(1 - \frac{1}{\mathcal{H}_Q(a'|s')})|Q(s',a')|1_{(a'\in\mathcal{U}(Q))}\}$.*

Before the proof, we redefine some notation to make the subsequent exposition clearer.

Denote state space $\mathcal{S}_\mathcal{D}$ be the state space defined by the distribution of dataset $\mathcal{D}$, $\mathcal{A}_\mathcal{D}$ is the action space defined by the dataset $\mathcal{D}$, and actions not belong to this space is the OOD actions. The optimal Q-value $Q^*$ on $\mathcal{S}_\mathcal{D} \times \mathcal{A}_\mathcal{D}$ can be derived by optimize the following Bellman equation:

$$\begin{aligned}
\mathcal{L}(Q) :=&\mathbb{E}_{s\sim P_\mathcal{D}(s),a\sim\pi(a|s)}[Q(s,a) - \mathcal{B}Q(s,a)]^2\\
=&\mathbb{E}_{s\sim P_\mathcal{D}(s),a'\sim\pi(a|s)}[Q(s,a) - (r(s,a) + \gamma\mathbb{E}_{s'\sim P_\mathcal{D}(s')}[\max_{a'} Q(s',a')])]^2,
\end{aligned} \tag{E.2}$$

where $(s,a) \in \mathcal{S}_\mathcal{D} \times \mathcal{A}_\mathcal{D}$, $Q^* = \mathcal{B}Q^*$.

We first introduced Lemma E.1 to facilitate the proof of Theorem 4.4.

**Lemma E.1.** *For any $s \in \mathcal{S}_\mathcal{D}, a \in \mathcal{A}_\mathcal{D}$, with probability $1 - \eta$,*

$$|\mathcal{B}Q^*(s,a) - \mathscr{F}Q^*(s,a)| \le \gamma(1 - \alpha)(1 - \beta)||Q^*(\cdot,\cdot)||_\infty, \tag{E.3}$$

*and with probability $\eta$,*

$$|\mathcal{B}Q^*(s,a) - \mathscr{F}Q^*(s,a)| \le \gamma(1 - \alpha)(1 - \frac{1}{\mathcal{H}_{Q^*}(a'|s')})||Q^*(\cdot,\cdot)||_\infty. \tag{E.4}$$

*Proof of Lemma E.1.*

Direct computation shows that

$$|\mathcal{B}Q(s,a) - \mathscr{F}Q(s,a)|$$

$$= |r(s,a) + \gamma\mathbb{E}_{s'\sim P_\mathcal{D}(s')}\{\max_{a'} Q(s',a')\} - r(s,a) - \gamma\mathbb{E}_{s'\sim P_\mathcal{D}(s')}\{\max_{a'}[\alpha Q(s',a') + (1-\alpha)Q_L(s',a')]\}|$$

$$\leq \gamma\mathbb{E}_{s'\sim P_\mathcal{D}(s')}|\max_{a'}\{Q(s',a') - [\alpha Q(s',a') + (1-\alpha)Q_L(s',a')]\}|$$

$$\leq \gamma\mathbb{E}_{s'\sim P_\mathcal{D}(s')}|\max_{a'}\{Q(s',a') - [\alpha Q(s',a') + (1-\alpha)Q_L(s',a')]\}|$$

$$\leq \gamma\mathbb{E}_{s'\sim P_\mathcal{D}(s')}\max_{a'}|Q(s',a') - [\alpha Q(s',a') + (1-\alpha)Q_L(s',a')]|$$

$$= \gamma\mathbb{E}_{s'\sim P_\mathcal{D}(s')}\max_{a'}|Q(s',a') - [(\alpha + (1-\alpha)\beta)Q(s',a')\mathbb{1}_{(a'\notin\mathcal{U}(Q))} + (\alpha + \frac{(1-\alpha)}{\mathcal{H}_Q(a'|s')})Q(s',a')\mathbb{1}_{(a'\in\mathcal{U}(Q))}]|$$

$$= \gamma\mathbb{E}_{s'\sim P_\mathcal{D}(s')}\max_{a'}|(1-\alpha - (1-\alpha)\beta)Q(s',a')\mathbb{1}_{(a'\notin\mathcal{U}(Q))} + (1-\alpha - \frac{(1-\alpha)}{\mathcal{H}_Q(a'|s')})Q(s',a')\mathbb{1}_{(a'\in\mathcal{U}(Q))}|$$

$$\leq \gamma\mathbb{E}_{s'\sim P_\mathcal{D}(s')}\max_{a'}\{(1-\alpha)(1-\beta))|Q(s',a')|\mathbb{1}_{(a'\notin\mathcal{U}(Q))} + (1-\alpha)(1-\frac{1}{\mathcal{H}_Q(a'|s')})|Q(s',a')|\mathbb{1}_{(a'\in\mathcal{U}(Q))}\}.$$

$$\text{(E.5)}$$

Then with probability $1 - \eta$,

$$|\mathcal{B}Q^*(s,a) - \mathscr{F}Q^*(s,a)| \leq \gamma\mathbb{E}_{s'\sim P_\mathcal{D}(s')}\max_{a'}\{(1-\alpha)(1-\beta))|Q^*(s',a')|\}$$

$$\leq \gamma(1-\alpha)(1-\beta)\|Q^*(\cdot,\cdot)\|_\infty, \qquad \text{(E.6)}$$

and with probability $\eta$,

$$|\mathcal{B}Q^*(s,a) - \mathscr{F}Q^*(s,a)| \leq \gamma\max_{s'}\max_{a'}\{(1-\alpha)(1-\frac{1}{\mathcal{H}_{Q^*}(a'|s')})|Q(s',a')|\}$$

$$\leq \gamma(1-\alpha)(1-\frac{1}{\mathcal{H}_{Q^*}(a'|s')})\|Q^*(\cdot,\cdot)\|_\infty. \qquad \text{(E.7)}$$

Then we finish the proof.

Next, we will give a brief proof of Theorem 4.4 with the results of Theorem 4.3 and Lemma E.1. Suppose the stationary point or the optimal Q-value derived based on the QDQ Bellman operator is $Q^\Delta$, which satisfying: $Q^\Delta = \mathscr{F}Q^\Delta$.

Then with probability $1 - \eta$ we have

$$\begin{aligned}
\left\|Q^\Delta - Q^*\right\|_\infty &= \left\|\mathscr{F}Q^\Delta - \mathscr{F}Q^* + \mathscr{F}Q^* - \mathcal{B}Q^*\right\|_\infty \\
&\leq \left\|\mathscr{F}Q^\Delta - \mathscr{F}Q^*\right\|_\infty + \left\|\mathscr{F}Q^* - \mathcal{B}Q^*\right\|_\infty \\
&\leq c\gamma\|Q^\Delta - Q^*\|_\infty + \gamma(1-\alpha)(1-\beta)\|Q^*(\cdot,\cdot)\|_\infty.
\end{aligned}$$

Hence,

$$\left\|Q^\Delta - Q^*\right\|_\infty \leq \gamma(1-\alpha)(1-\beta)(1-c\gamma)^{-1}\|Q^*(\cdot,\cdot)\|_\infty.$$

Set $\epsilon = \gamma(1-\alpha)(1-\beta)(1-c\gamma)^{-1}\|Q^*(\cdot,\cdot)\|_\infty$ finish the proof.

*Remark* E.1. The error $\epsilon = \gamma(1-\alpha)(1-\beta)(1-c\gamma)^{-1}\|Q^*(\cdot,\cdot)\|_\infty$ can be 0 by setting $\beta = 1$. In practice, we can ensure that $\epsilon$ converges to a small value by appropriately adjusting the parameters $\alpha$ and $\beta$.

*Remark* E.2. Our primary focus lies on the scenario where $\max_{a'}\{(1-\beta))|Q(s',a')|\mathbb{1}_{(a'\notin\mathcal{U}(Q))} + (1-\frac{1}{\mathcal{H}_Q(a'|s')})|Q(s',a')|\mathbb{1}_{(a'\in\mathcal{U}(Q))}\} = \max_{a'}\{(1-\beta))|Q(s',a')|\mathbb{1}_{(a'\notin\mathcal{U}(Q))}\}$. This preference stems from the potential minuteness of $\eta$. Firstly, we can rely on Theorem 1 in [20] to ensure that the consistency model can converge to ground truth, thus guaranteeing the fidelity of the learned Q-distribution. Consequently, the accuracy of our uncertainty estimation is upheld, enabling us to effectively assess OOD points and pessimistically adjust the Q-value for such occurrences [4]. Secondly, in order to mitigate segmentation errors of the uncertainty set, we introduce the penalty factor $\beta$ in Eq.8. Finally, since actions in the set $\mathcal{U}(Q)$ are always penalized, the Q-value always takes a lower value when $a' \in \mathcal{U}(Q)$. Collectively, these measures reinforce our aim to maintain a small $\eta$.

Theorem 4.4 shows the optimal Q-value by QDQ can closely approximate the true optimal Q-value $Q^*$ over $\mathcal{S}_\mathcal{D} \times \mathcal{A}_\mathcal{D}$. Then we also provide the following corollary to show that substitute the optimal Bellman operator $\mathcal{B}Q$ with $\mathscr{F}Q$ will introduce controllable error at each step, and give an step-wise analysis of the convergence of QDQ operator $\mathscr{F}Q$.

Let $\zeta_k(s,a) = |Q^k(s,a) - Q^*(s,a)|$ be the total estimation error of Q-value learned by QDQ algorithm and the optimal in-distribution Q-value at step k of the value iteration. Let $\delta_k(s,a) = |Q^k(s,a) - \mathscr{F}Q^k(s,a)|$ be the Bellman residual induced by QDQ Bellman operator $\mathscr{F}Q$ at step k. Assume $\delta_k^*(s,a) = |Q^k(s,a) - \mathcal{B}Q^k(s,a)|$ be the Bellman residual for the optimal in-distribution Q-value optimization.

**Corollary 1.** *At step k of value iteration, substitute the optimal Bellman operator $\mathcal{B}Q$ with $\mathscr{F}Q$ introduce an arbitrary small error as $\xi$:*
$$\zeta_k(s,a) \leq \delta_k^*(s,a) + \xi + \gamma\mathbb{E}_{s'\sim D}\max_{a'}|\zeta_{k-1}(s,a)|$$

Then we introduce our second Lemma E.2 to help the proof of Corollary 1:

**Lemma E.2.** *For any $s \in \mathcal{S}_\mathcal{D}, a \in \mathcal{A}_\mathcal{D}$, with probability $1-\eta$,*
$$\delta_k(s,a) \leq \delta_k^*(s,a) + \gamma(1-\alpha)(1-\beta)||Q(s',a')||_\infty. \tag{E.8}$$
*With probability $\eta$,*
$$\delta_k(s,a) \leq \delta_k^*(s,a) + \gamma(1-\alpha)(1-\frac{1}{\mathcal{H}_Q(a'|s')})||Q(s',a')||_\infty. \tag{E.9}$$

*Proof of Lemma E.2.*
$$\begin{aligned}
\delta_k(s,a) =& |Q^k(s,a) - \mathscr{F}Q^k(s,a)| \\
=& |Q^k(s,a) - \mathcal{B}Q^k(s,a) + \mathcal{B}Q^k(s,a) - \mathscr{F}Q^k(s,a)| \\
\leq& \delta_k^*(s,a) + |\mathcal{B}Q(s,a) - \mathscr{F}Q(s,a)|\delta_k^*(s,a) + \gamma(1-\alpha)(1-\beta)||Q(s',a')||_\infty. \quad \text{(E.10)}
\end{aligned}$$
By Lemma E.1, with probability $1-\eta$,
$$\begin{aligned}
\delta_k(s,a) =& |Q^k(s,a) - \mathscr{F}Q^k(s,a)| \\
=& |Q^k(s,a) - \mathcal{B}Q^k(s,a) + \mathcal{B}Q^k(s,a) - \mathscr{F}Q^k(s,a)| \\
\leq& \delta_k^*(s,a) + |\mathcal{B}Q(s,a) - \mathscr{F}Q(s,a)| \\
\leq& \delta_k^*(s,a) + \gamma(1-\alpha)(1-\beta)||Q(s',a')||_\infty. \quad \text{(E.11)}
\end{aligned}$$
With probability $\eta$,
$$\begin{aligned}
\delta_k(s,a) =& |Q^k(s,a) - \mathscr{F}Q^k(s,a)| \\
=& |Q^k(s,a) - \mathcal{B}Q^k(s,a) + \mathcal{B}Q^k(s,a) - \mathscr{F}Q^k(s,a)| \\
\leq& \delta_k^*(s,a) + |\mathcal{B}Q(s,a) - \mathscr{F}Q(s,a)| \\
\leq& \delta_k^*(s,a) + \gamma(1-\alpha)(1-\frac{1}{\mathcal{H}_Q(a'|s')})||Q(s',a')||_\infty. \quad \text{(E.12)}
\end{aligned}$$

Then we finish the proof.

Next, we will give a brief proof of Corollary 1.
$$\begin{aligned}
\zeta_k(s,a) =& |Q^k(s,a) - Q^*(s,a)| \\
=& |Q^k(s,a) - \mathscr{F}Q^{k-1}(s,a) + \mathscr{F}Q^{k-1}(s,a) - Q^*(s,a)| \\
\leq& |Q^k(s,a) - \mathscr{F}Q^{k-1}(s,a)| + |\mathscr{F}Q^{k-1}(s,a) - Q^*(s,a)| \\
=& \delta_k(s,a) + |\mathscr{F}Q^{k-1}(s,a) - Q^*(s,a)| \\
=& \delta_k(s,a) + |\mathscr{F}Q^{k-1}(s,a) - \mathcal{B}Q^{k-1}(s,a) + \mathcal{B}Q^{k-1}(s,a) - \mathcal{B}Q^*(s,a)| \\
\leq& \delta_k(s,a) + |\mathscr{F}Q^{k-1}(s,a) - \mathcal{B}Q^{k-1}(s,a)| + |\mathcal{B}Q^{k-1}(s,a) - \mathcal{B}Q^*(s,a)| \\
=& \delta_k(s,a) + |\mathscr{F}Q^{k-1}(s,a) - \mathcal{B}Q^{k-1}(s,a)| + |\gamma\mathbb{E}_{s'\sim P_D}(s')\max_{a'}Q^{k-1}(s,a) - \gamma\mathbb{E}_{s'\sim P_D}(s')\max_{a'}Q^*(s,a)| \\
\leq& \delta_k(s,a) + |\mathscr{F}Q^{k-1}(s,a) - \mathcal{B}Q^{k-1}(s,a)| + \gamma\mathbb{E}_{s'\sim D}\max_{a'}|Q^{k-1}(s,a) - Q^*(s,a)| \\
=& \delta_k^*(s,a) + \gamma(1-\alpha)(1-\frac{1}{\mathcal{H}_Q(a'|s')})||Q(s',a')||_\infty + \gamma\mathbb{E}_{s'\sim D}\max_{a'}|\zeta_{k-1}(s,a)| \quad \text{(E.13)}
\end{aligned}$$

By Lemma E.1 and Lemma E.2, we can easily derive, with probability $1 - \eta$,

$$\zeta_k(s,a) \leq \delta_k^*(s,a) + \gamma(1-\alpha)(1-\beta)||Q(s',a')||_\infty + \gamma\mathbb{E}_{s'\sim D}\max_{a'}|\zeta_{k-1}(s,a)| \qquad \text{(E.14)}$$

Set $\xi = \gamma(1-\alpha)(1-\beta)||Q(s',a')||_\infty$, then we finish the proof.

*Remark* E.3. During the value iteration, there are two error accumulate: $\delta_k^*(s,a)$ and $\xi$. Given the convergence of optimal Q-value iteration, $\delta_k^*(s,a)$ tends to approach an arbitrarily small value and may even converge to 0 under optimal circumstances. The error $\xi = \gamma(1-\alpha)(1-\beta)||Q(s',a')||_\infty$ induced by using the Bellman operator $\mathscr{F}Q$ instead of the optimal Bellman operator can also approach 0 as discussed in Remark E.1.

# F    Gap expanding of the QDQ algorithm.

In this section we will show that QDQ also has a Gap expanding property as discussed in CQL [3].

**Proposition F.1.** *The QDQ algorithm is Gap expanding for Q-values within-distribution and Q-values out of distribution.*

(1) If $a \notin \mathcal{U}(Q)$ and is indeed in-distribution action, the target Q-value for the in-distribution point is $(\alpha + (1-\alpha)\beta)Q$. The target Q-value for the OOD point is $(\alpha + \frac{(1-\alpha)}{\mathcal{H}_Q(a|s)})Q$. And $(\alpha + (1-\alpha)\beta)Q - (\alpha + \frac{(1-\alpha)}{\mathcal{H}_Q(a|s)})Q > 0$ definitely. And this suggest Q-value will prefer in-distribution actions.

(2) If $a \notin \mathcal{U}(Q)$ and is indeed OOD action. In such cases, the penalty parameter $\beta$ is introduced to penalize the Q-value, resulting in $(\alpha + (1-\alpha)\beta)Q < Q$. Consequently, misclassifications of OOD actions can be handled in a pessimistic manner.

While introducing the penalty parameter $\beta$ may slightly slow down optimization updates for true in-distribution actions, it is crucial to prioritize control over out-of-distribution (OOD) actions due to the potentially severe consequences of exploration errors. In balancing the optimization of Q-learning with a pessimistic approach to OOD actions, the focus should be on reducing the impact of OOD actions. In fact, compared to previous pessimistic Q-value methods, QDQ offers greater flexibility in managing these values. This includes incorporating an uncertainty-aware learning objective and adjusting Q-values based on their uncertainty.

# G    Experiment Details and More Results

## G.1    Real Q dataset generation.

In Section 3.1, we use a $k$-step sliding window approach with a length of $\mathcal{T}$ to traverse each trajectory in the dataset $\mathcal{D}$ and generate the Q-value dataset $\mathcal{D}_Q$ based on Equation 5. A good Q-value dataset for uncertainty estimation should cover a broad state and action space to accurately characterize the Q distribution and detect high uncertainty in the OOD action region. This coverage helps identify actions with significant Q-value uncertainty in OOD areas.

Choosing the right value for the sliding step $k$ and the window length $\mathcal{T}$ is crucial. A large value for $k$ may result in a smaller Q dataset, while a small value may lead to excessive homogenization of the Q dataset. Both situations can negatively impact the learning of the Q-value distribution. If the Q-value dataset is too small, the learned Q distribution may not generalize well across the state-action space $\mathcal{S} \times \mathcal{A}$. Conversely, if the Q-values are too homogeneous, it can hinder feature learning by the distribution learner.

To illustrate this, Figure G.1 shows the Q dataset distributions obtained for different values of $k$ using the *halfcheetah-medium* dataset. When $k = 1$, the Q distribution is more concentrated, resulting in more homogeneous Q-values. In contrast, with $k = 50$, the Q distribution becomes sparser, showing a greater inclination towards individual features.

In the experiments, we consider various factors when setting the value of $k$, including the trajectory length, the width of the sliding window, and the derived Q-value dataset size. Throughout all experiments, we set $k$ to 10. Interestingly, we observed that the distribution of Q-values remains

robust to minor adjustments in $k$, indicating that the choice of $k$ does not necessitate overly stringent tuning.

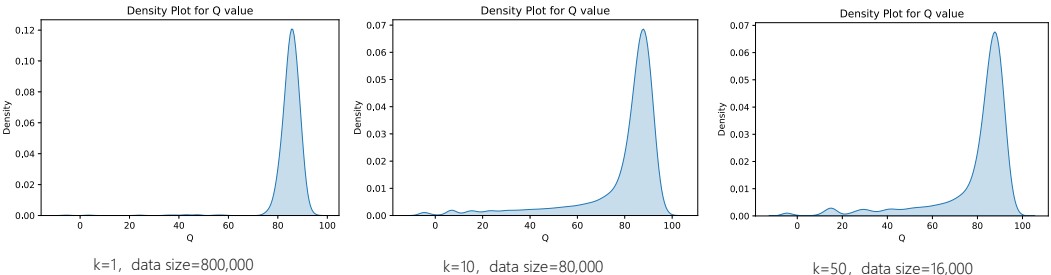

Figure G.1: The derived Q-value distribution when using difference sliding step and same window width to scan over the trajectory's on *halfcheetah-medium* dataset.The width of the sliding window is set to 200. The Q-value is scaled to facilitate comparison.

When choosing the width of the sliding window, $\mathcal{T}$, we must consider factors such as trajectory length and the resulting size of the Q-value dataset. When $\mathcal{T}$ increases, the size of the Q-value dataset decreases. However, if $\mathcal{T}$ is too small, it might truncate essential information from the true Q-value. We give the experimental analysis of Q-value distribution for different $\mathcal{T}$ on the sparse reward task Antmaze-medium-play in Figure G.2.

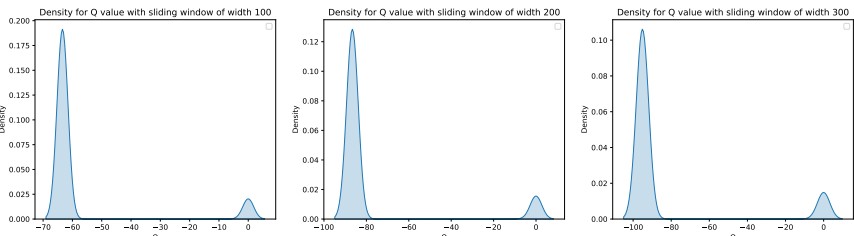

Figure G.2: The Q distribution of the Antmaze-medium-play dataset with varying sliding window widths (100 to 300 steps) is shown in the figure. Widening the sliding window does not change the shape of the Q distribution, even though a larger window covers more information for this sparse reward task with many short trajectories. Instead, enlarging the sliding window decreases the Q value and compresses the size of the derived Q data.

In our experiments, we opted for $\mathcal{T} = 200$ across all tasks. This choice considers factors such as the maximum trajectory length (1000), the decay rate of $\gamma^{m-1}$ in Eq.5. Based on the analysis provided in Section B, $\mathcal{T}$ does not need to be very large. We also found that minor adjustments to $\mathcal{T}$ do not significantly affect the Q-value distribution, indicating that strict tuning of this parameter is unnecessary.

## G.2 The distribution of Q-value function.

The Q-distributions based on the truncated Q-value and the learned distribution from the consistency model are shown in Figure G.3 for Gym-MuJoCo tasks and in Figure G.4 for Antmaze tasks. In Figure G.3, the consistency model roughly captures the main characteristics of the true Q-value distribution. However, for the Antmaze task (Figure G.4), the learned distribution shows some fluctuations and a slightly wider support compared to the true Q-value distribution, particularly in the dynamic goal task ("-diverse" task).

From the distribution of Antmaze we observe that trajectories for these tasks mainly consist of suboptimal or failed experiences, this underscores the challenging nature of the Antmaze task.

Further the narrower data distribution make it easier to take OOD actions during offline training and fewer positive experiences limits the optimisation of Q, these all potentially leading to failure of these kinds of tasks.

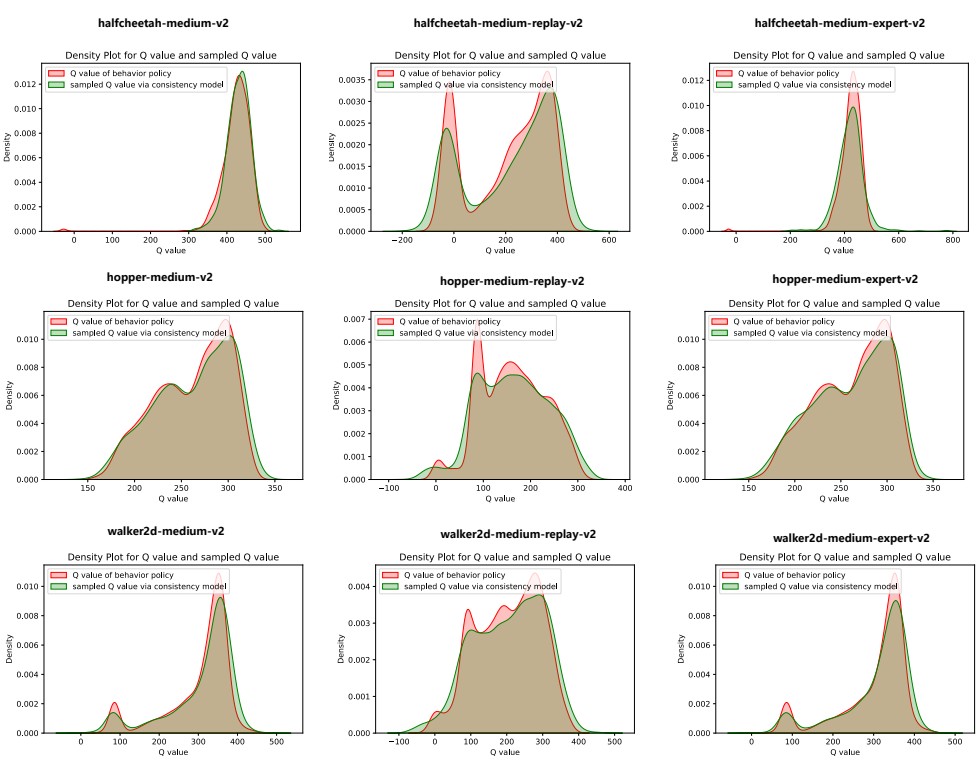

Figure G.3: The Q-value distribution based on the truncated Q-value v.s. the sample Q-value distribution via the learned consistency model for Gym-MuJoCo tasks.

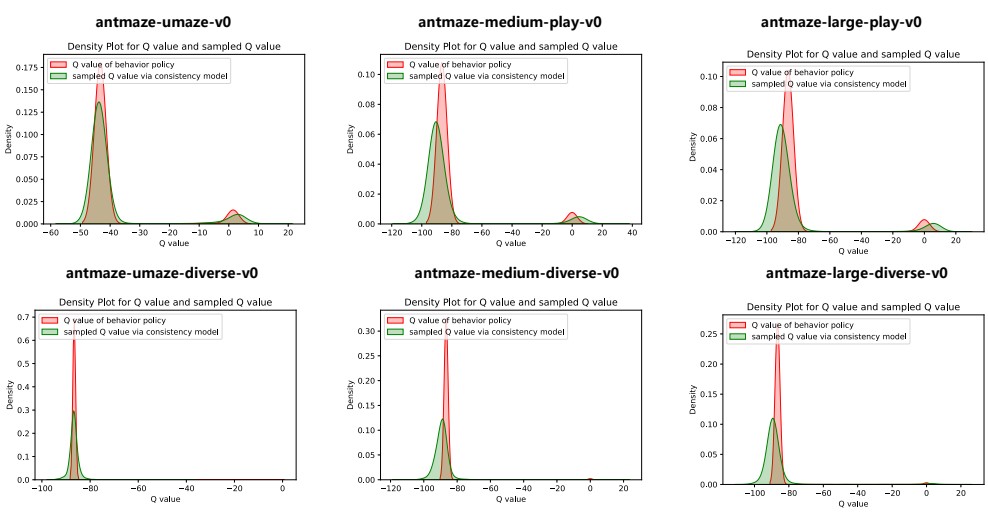

Figure G.4: The Q-value distribution based on the truncated Q-value v.s. the sample Q-value distribution via the learned consistency model for Antmaze tasks.

## G.3 Efficiency of the uncertainty measure.

The uncertainty measure is crucial for guiding the Q-value towards a pessimistic regime within the QDQ algorithm. In this section, we will first verify how uncertainty can assess the overestimation of Q-values in OOD regions. Then, we will discuss how the uncertainty set $\mathcal{U}(Q)$ can be shaped using the hyperparameter $\beta$.

To understand the differences in uncertainty between in-distribution actions and OOD actions from a random policy, we can compare their distributions. In the left graph of Figure G.5, the red line represents the 95% quantile of the standard deviation of sampled Q-values from the learned Q-distribution for in-distribution actions. In the right graph, this 95% quantile corresponds to approximately the 75% quantile of the standard deviation for OOD actions. This indicates that OOD actions contribute to a heavy-tailed distribution of Q-value uncertainty, resulting in larger values compared to in-distribution actions.

Figure G.6 further illustrates this, showing that the standard deviation of sampled Q-values from the learned policy sharply increases when Q-values are overestimated and become uncontrollable. These observations suggest that the uncertainty measure in the QDQ algorithm effectively captures the overestimation phenomenon in OOD actions.

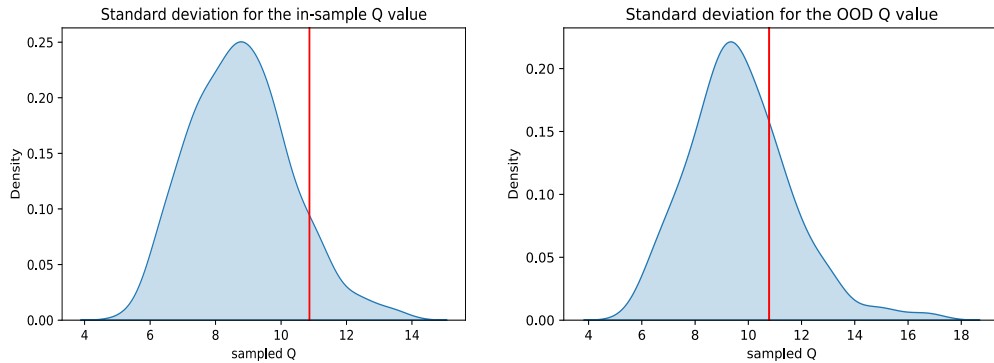

Figure G.5:  The uncertainty distribution of Q-value based on in-distribution action v.s. the uncertainty distribution of Q-value based on OOD actions(by a ramodm policy).

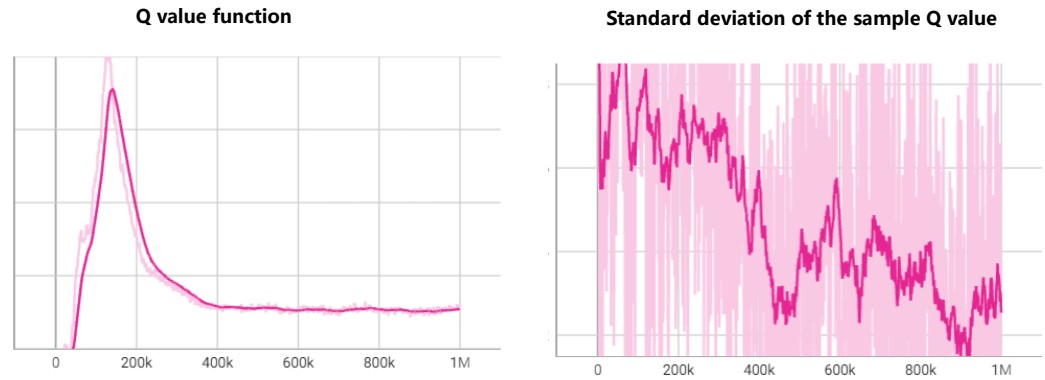

Figure G.6:  The Q value learned by QDQ(left), and the standard deviation of the sample Q-value from the consistency model for same state and action pair.

Table G.1: The hyperparameter used in consistency model training.

| | Hyperparameter | Value |
|---|---|---|
| Shared parameter | Batch size | 256 |
| | Time embedding | Gaussian Fourier features embeddings |
| | Fourier scale factor | 16 |
| | Time embedding size | 8 |
| | Minimum time | 0.002 |
| | Maximum time | 80 |
| | Group norm | True |
| | Activation in the hidden layer | Swish |
| | Latent dim | 256 |
| | Hidden Layer | 3 |
| | sample size | 50 |
| Diffusion model | Optimizer | Adam |
| | Learning rate | 5e-4 |
| | SDE | KVESDE [68] |
| | sampling method | heun sampler |
| | Number of iterations | 80000 |
| Consistency model | Optimizer | RAdam |
| | Learning rate | 4e-4 |
| | trajectory interval | 1000 |
| | trajectory interval factor as in [68] | 7 |
| | consistency loss norm | l2 |
| | SDE solver | heun |
| | sampling method | one-step sampler |
| | Number of iterations | 160000 |

The uncertainty set $\mathcal{U}(Q)$ is derived using the upper $\beta$-quantile of the entire uncertainty value of Q over the action taken by the learning policy. In Section 5.2, we provide a brief discussion on determining the appropriate $\beta$ during experiments. A higher $\beta$ may allow for a more generous attitude towards OOD actions, which is suitable for tasks with a wide data distribution or that are robust for action change. Conversely, a lower $\beta$ suggests more rigid control over OOD actions, appropriate for tasks with a narrow data distribution or that are sensitive. During experiments, a rough starting point for setting $\beta$ can be obtained by comparing the quantiles of the uncertainty distribution based on in-distribution actions and OOD actions. For instance, as shown in Figure G.4, parameter tuning might begin with $\beta = 0.75$ or $\beta = 0.80$.

### G.4 Implementation details for QDQ algorithm.

Implementation of QDQ algorithm contains consistency distillation for the consistency model and offline RL training. The whole algorithm is implemented with jax [67]. The training process of consistency distillation and offline RL is independent, we first train the consistency model by consistency distillation and save the converged model. Then we use the pretrained consistency model as the Q-value distribution sampler and go through the offline RL training, see Algorithm 1 for the whole training process.

**Consistency Distillation.** The consistency model is trained using a pretrained diffusion model [33]. The training process follows the official implementation [4] of the consistency model [20]. Since the consistency model is designed for image data, we modified the initial architecture, the *NCSN++ model* [33], to better fit offline RL data. For instance, we replaced the U-Net architecture with a multilayer perceptron (MLP) that has three hidden layers, each with 256 units. This simplified architecture is used to learn the consistency function $f_\theta(\boldsymbol{x_t}, t)$. The main hyperparameters for the consistency distillation are shown in Table G.1, covering both diffusion model training and consistency model training.

---

[4]https://github.com/openai/consistency_models_cifar10

Table G.2: The hyperparameters used in Actor-Critic training.

| | Hyperparameter | Value |
|---|---|---|
| TD3 training | Optimizer | Adam |
| | Batch size | 256 |
| | Discount factor | 0.99 |
| | Actor learning rate | 3e-4 |
| | Critic learning rate | 3e-4 |
| | Number of iterations | 1e6 |
| | Target update rate $\tau$ | 0.005 |
| | Actor noise | 0.2 |
| | Actor noise clipping | 0.5 |
| | Actor update frequency | 2 |
| | Actor activation | relu |
| | Critic activation | mish |
| | Evaluation episode length | 10 for MuJoCo |
| | | 100 for Antmaze |
| Architecture | Actor hidden dim | 256 |
| | Actor layers | 3 |
| | Critic hidden dim | 256 |
| | Critic layers | 3 |
| QDQ specific | sliding window step k | 10 |
| | sliding window width $\mathcal{T}$ | 200 |
| | uncertainty parameter $\beta$ | 0.8 for hopper task |
| | | 0.9 for other task |
| | pessimistic parameter $\alpha$ | 0.99 for halfcheetah ,hoper-medium and hoper-medium-replay, Antmaze task |
| | | 0.95 for other task |
| | entropy parameter $\gamma$ | 0 for halfcheetah task except the expert dataset |
| | | 1 for other task |

**Offline RL training.** The offline RL training follows TD3 [46], which has a delayed update schedule for both the target Q network and the target policy network. For the Gym-MuJoCo tasks, we use the raw dataset without any preprocessing, such as state normalization or reward adjustment. For the AntMaze tasks, we apply the same reward tuning as IQL [10], with no additional preprocessing for the state. The hyperparameters used in offline RL training are shown in Table G.2.

## G.5 Learning Curve

The learning curve of Gym-Mujoco tasks are shown in Figure G.7. The learning curve of AntMaze tasks are shown in Figure G.8.

## G.6 Computation efficiency of QDQ

We provide a detailed discussion of the computational efficiency of the QDQ algorithm we proposed, focusing on the training cost of the consistency model and computation coefficiency of QDQ (the distribution-based bootstrap method) compared with SOTA uncertainty estimation methods based on Q-value ensembles.

Regarding the training cost of the consistency model, we believe it is nearly negligible. Training a diffusion model on a 4090 GPU takes about 5.2 minutes, while training a consistency model using this pretrained diffusion model takes around 16 minutes. Additionally, the consistency model can be stored and reused for subsequent RL experiments, eliminating the need for retraining.

For a comparison of the computational costs of QDQ with SOTA uncertainty estimation methods in offline RL, see Table G.6. As mentioned earlier, QDQ achieves a significantly faster training speed compared to the ensemble-based uncertainty estimation method EDAC and other SOTA methods. The results for other methods are taken from Table 3 of EDAC [9].

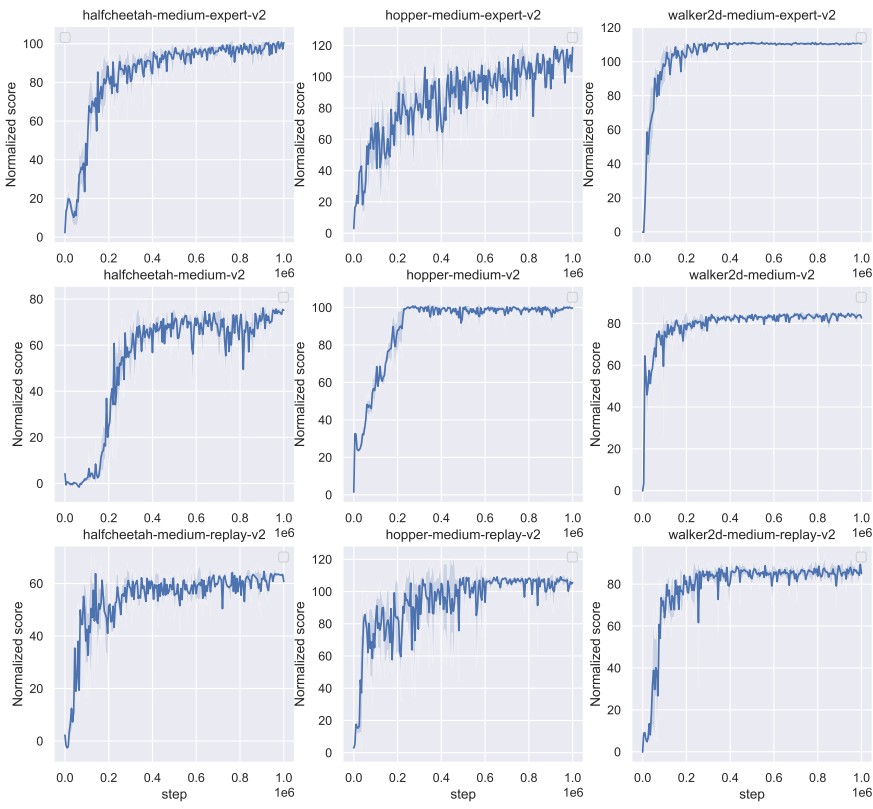

Figure G.7:  Training curve of different Mujoco Tasks. All results are averaged across 5 random seeds. The evaluation interval is 5000 with evaluation episode length 10.

Table G.3: Computational performance of QDQ and other SOTA methods.

|      | Runtime(s/epoch) | GPU Mem.(GB) |
|------|------------------|--------------|
| SAC  | 21.4             | 1.3          |
| CQL  | 38.2             | 1.4          |
| EDAC | 30.8             | 1.8          |
| **QDQ**  | **0.028**    | **0.74**     |

## G.7   Ablations

Although QDQ has three hyperparameters ($\alpha$, $\beta$, and $\gamma$) for flexibility, the tuning process is straightforward. For example, as discussed in Theorem 4.4 (Appendix E), theoretically, $(1-\alpha)(1-\beta)$ should be small. Since $\beta$ controls the size of the uncertainty set and requires flexibility across different tasks, we typically set $\alpha$ close to 1, tuning it between 0.9 and 0.995. This only requires a few experiments to find the optimal value. Our tuning process involves sequentially fixing parameters while selecting the best $\alpha$, then $\beta$, and finally $\gamma$. Based on the characteristics of different datasets, we can set each parameter to an initial value close to its optimal value. QDQ offers evidence-based guidelines for

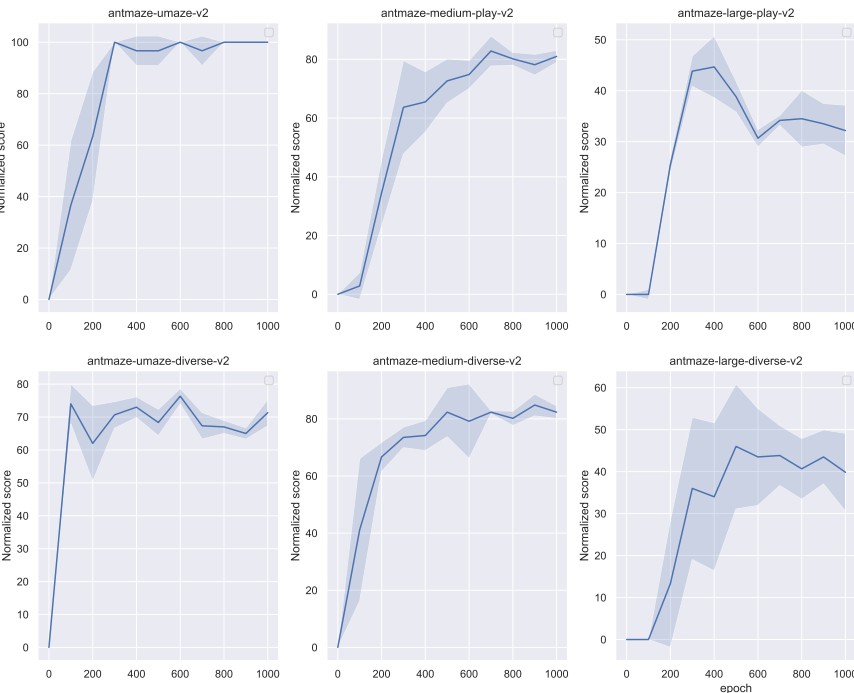

Figure G.8: Training curve of different Antmaze Tasks. All results are averaged across 5 random seeds. The evaluation interval is 100000 with evaluation episode length 100.

hyperparameter ranges, making tuning manageable. Additionally, QDQ's efficiency (see Table G.6) reduces the tuning burden.

In the ablation study, we choose four task which represent different kinds of dataset we analyzed during parameter study:

(1) Wide distribution: *halfcheetah-medium-v2*.

(2) Task sensitive: *hopper-medium-v2*.

(3) Demonstration: *walker2d-medium-expert-v2*.

(4) Narrow distribution: *umaze-diverse-v0*.

We perform ablation study for parameters we discussed in Section 5.2. We compare the performances of three different settings for the three parameters respectively.

The learning curves for four types of tasks with different uncertainty-aware weights $\alpha$ (Eq. 7) are shown in Figure G.9. For the *halfcheetah-medium-v2* and *hopper-medium-v2* datasets, decreasing $\alpha$ harms performance. For the *walker2d-medium-expert-v2* dataset, a slightly lower $\alpha$ increases training volatility. In contrast, the *umaze-diverse-v0* dataset shows high sensitivity to $\alpha$. This sensitivity occurs because the Antmaze task requires combining different suboptimal trajectories, which challenges algorithms like QDQ that are not fully in-sample but in-support. For instance, an $\alpha$ value that is too high or too low can lead to overly optimistic or pessimistic Q-values, causing the algorithm to favor actions that do not align with the exact suboptimal trajectories.

The ablation study of the uncertainty-related parameter $\beta$ is presented in Figure G.11. For the wide dataset *halfcheetah-medium-v2*, a higher $\beta$ is preferred, indicating less control over the uncertainty

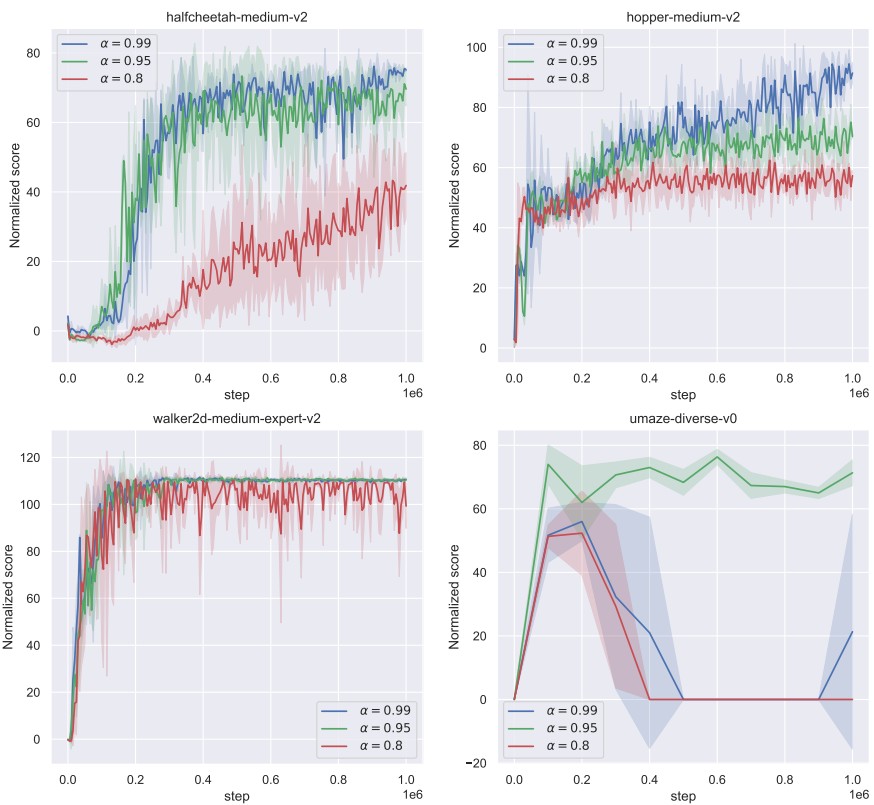

Figure G.9: Training curve of four different Tasks when using different $\alpha$ in Eq. 7. All results are averaged across 5 random seeds. The evaluation interval is 5000 with evaluation episode length 10 for Mujoco tasks. The evaluation interval is 100000 with evaluation episode length 100 for Antmaze task.

penalty and a smaller uncertainty set. In contrast, the performance for the *hopper-medium-v2* and *umaze-diverse-v0* datasets is sensitive to small changes in $\beta$ due to their task sensitivity and narrow distribution. The *walker2d-medium-expert-v2* dataset, however, is robust to small changes in $\beta$.

We present the performance of different tasks for the entropy parameter $\gamma$ in Figure G.12. For the wide distribution in *halfcheetah-medium-v2*, a larger $\gamma$ negatively impacts performance. The *hopper-medium-v2* and *umaze-diverse-v0* datasets also show sensitivity to the parameter $\gamma$. In the *walker2d-medium-expert-v2* dataset, a very small $\gamma$ may introduce volatility into the training process. However, the final result's convergence is not significantly affected.

Furthermore, as discussed in Section 5.2, the gamma term primarily stabilizes the learning of a simple Gaussian policy, especially for action-sensitive and dataset-narrow tasks. We note that Gaussian policies are prone to sampling risky actions because they can only fit a single-mode policy. To verify the impact of uncertainty-aware Q-value optimization in QDQ, we compared the performance of Q-values without uncertainty control (using Bellman optimization like in online RL settings) to the QDQ algorithm on the action-sensitive hopper-medium dataset, using identical gamma settings. Figure G.10 shows that introducing an uncertainty-based constraint for the Q-value function in QDQ significantly improves training stability, convergence speed, and overall performance. This supports

the effectiveness of QDQ's uncertainty-aware Q-value optimization. We believe that a stronger learning policy will further reduce the need for this stabilizing term.

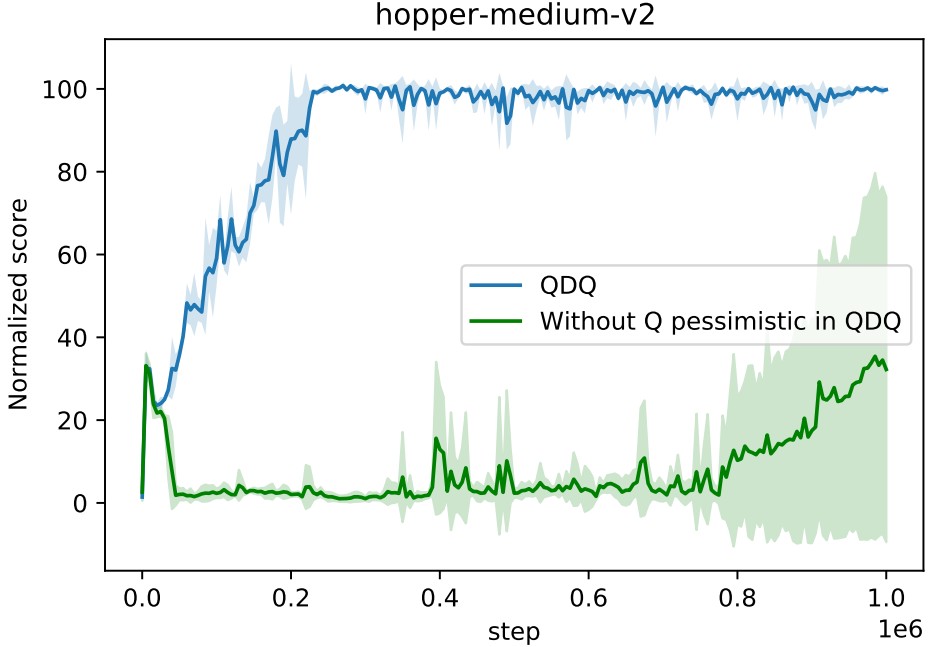

Figure G.10: The training curve of the hopper-medium dataset with QDQ's uncertainty pessimistic Q learning and without Q-value adjustments is shown. The green curve indicates that the $\gamma$ term in Eq. 9 has a limited impact on performance. Comparing the learning curves, QDQ's uncertainty pessimistic Q learning boosts performance, leading to faster convergence and greater stability.

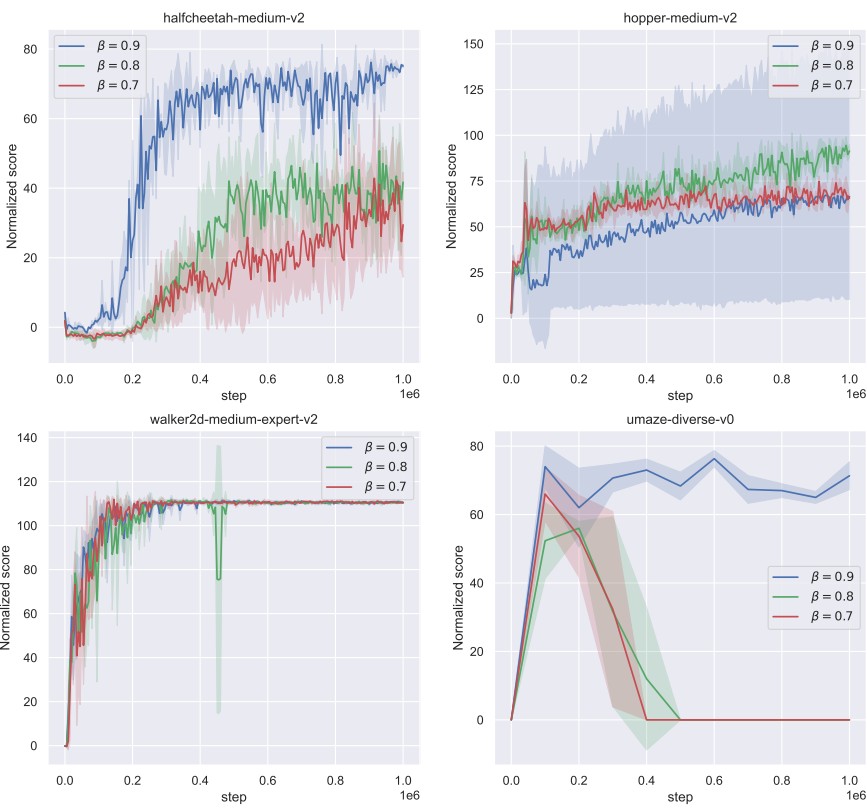

Figure G.11: Training curve of four different Tasks when using different $\beta$ in Section 3.2. All results are averaged across 5 random seeds. The evaluation interval is 5000 with evaluation episode length 10 for Mujoco tasks. The evaluation interval is 100000 with evaluation episode length 100 for Antmaze task.

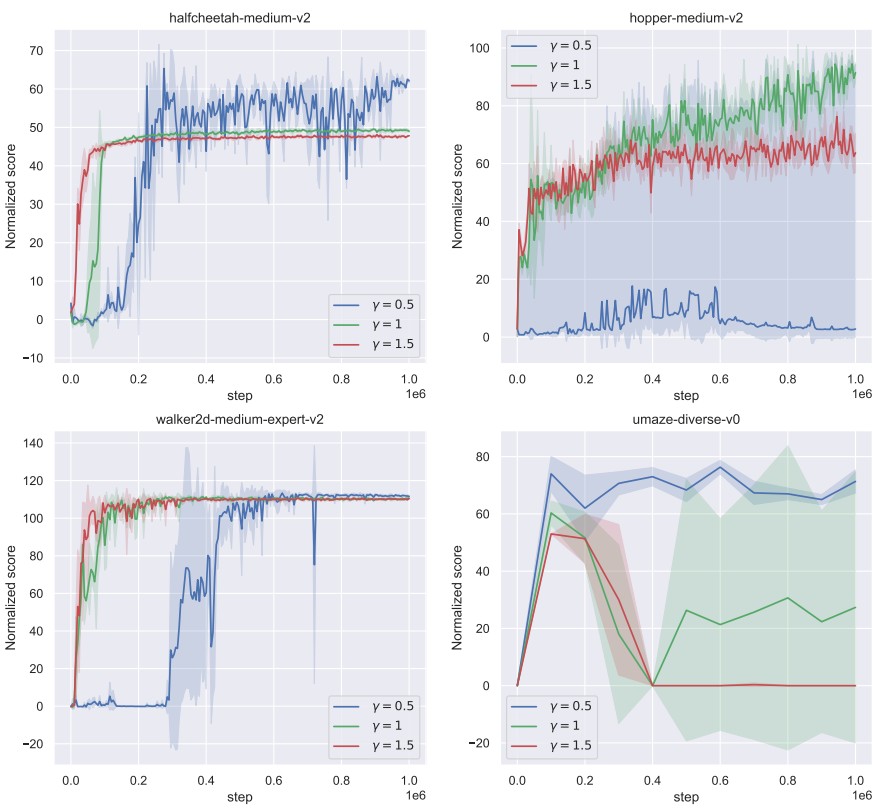

Figure G.12: Training curve of four different Tasks when using different $\gamma$ in Eq. 9. All results are averaged across 5 random seeds. The evaluation interval is 5000 with evaluation episode length 10 for Mujoco tasks. The evaluation interval is 100000 with evaluation episode length 100 for Antmaze task.

