# OpenReview forum: "Q-Distribution guided Q-learning for offline reinforcement learning: Uncertainty penalized Q-value via consistency model"
_NeurIPS.cc/2024/Conference — NeurIPS 2024 poster_

### Official Review · Reviewer_T4W8 · 2024-07-09

**Soundness:** 3
**Presentation:** 3
**Contribution:** 3
**Rating:** 6
**Confidence:** 3

**Summary:**

This paper addresses the well-known issue of OOD (Out-of-Distribution) actions in offline reinforcement learning by proposing the QDQ method, which penalizes the Q-values in regions with high uncertainty. To better estimate uncertainty, QDQ first constructs a truncated Q-value dataset for the behavior policy and learns the distribution of Q-values induced by the behavior policy using a consistency model, allowing subsequent estimation of the uncertainty for each state-action pair through one-step sampling. With this uncertainty estimation method, QDQ can perform  pessimistic value estimation for samples, thus avoiding the overestimation of Q-values for OOD actions. The paper provides a series of theoretical results to support the rationality of the QDQ method and demonstrates the advantages of the QDQ method through a series of experiments.

**Strengths:**

1. This paper proposes an innovative approach to estimate the sample uncertainty in the dataset by learning a consistency model and employing one-step sampling. This scheme is novel as it does not require maintaining multiple Q-networks to estimate uncertainty, as traditional methods do, thus incurring less computational overhead. Moreover, unlike the diffusion model, which requires multi-step denoising to obtain uncertainty, this approach ensures higher fidelity in uncertainty estimation without the need for iterative processes. Consequently, this scheme is practical in terms of implementation.

2. The paper has a solid theoretical foundation, demonstrating the feasibility and convergence of the algorithm.

3. The paper is well-written with clear expression, distinct structure, and logical flow.

**Weaknesses:**

1. First and foremost, the fundamental assumption of this paper is that a consistency model can learn a distribution of Q-values with high fidelity. However, why do Q-values need to satisfy the property of consistency? If we are learning the distribution of trajectories, it is reasonable to require the trajectory distribution to be consistent, but is this assumption also reasonable for the distribution of Q-values? This is the starting point of the paper, yet it does not argue for its necessity. Therefore, I suspect that for any type of generative model, even Variational Autoencoders (VAEs), the uncertainty of Q(s,a) could be calculated, thereby allowing the subsequent methods of this paper to be applied for learning.

2. Based on my understanding, Equation (6) actually calculates the emperical version $V_{\epsilon}(s,a)$ of uncertainty $Var(Q^{\pi_{\beta}})$  corresponding to the behavior policy $\pi_{\beta}$, rather than the uncertainty $Var(Q^{\pi})$ of  corresponding to the current policy $\pi$. However, it seems to me that in the subsequent use of $V_{\epsilon}(s,a)$ and in the proof of the theorems, $V_{\epsilon}(s,a)$ is directly used as $Var(Q^{\pi})$, which could be mathematically problematic. The authors need to provide an explanation for this point. If indeed $Var(Q^{\pi_{\beta}})$ is used in place of $Var(Q^{\pi})$, please justify the rationale.

3. This paper lacks some comparisons with related work. It could be compared with methods that use ensembles to estimate uncertainty [1] and methods that make pessimistic estimates for Out-of-Distribution (OOD) actions [2].


[1] Pessimistic Bootstrapping for Uncertainty-Driven Offline Reinforcement Learning.
[2] Supported value regularization for offline reinforcement learning.

**Questions:**

The same as Weaknesses

**Limitations:**

This paper does not mention any potential negative impacts that may arise from its work. It is recommended that this be supplemented.

---

> ### Author Rebuttal · Authors · 2024-08-07
>
> For weakness:
>
> 1.	We apologize for any ambiguity caused by our description of the consistency model. Consistency is a feature of the consistency model, not a requirement for the Q-value. The consistency model ensures a consistent relationship between the prior sample and the target sample during generation. This consistency is beneficial for computing Q-value uncertainty, as analyzed in Theorem 4.2, which shows that the Q-value distribution learned with the consistency model can ensure the absolute effect of action changes on the variance of the final bootstrap sample. This makes Q-value uncertainty more sensitive to OOD actions compared to the diffusion model. Additionally, the fast-sampling process of the consistency model significantly enhances QDQ's efficiency. Although one-step sampling may slightly reduce sample quality, QDQ only calculates uncertainty by the variance of the bootstrap sample, so this loss is negligible. Overall, the consistency model is highly suitable for uncertainty estimation due to its consistency, high fidelity, easier training, and faster sampling.
> Regarding the use of VAE and GAN for learning Q-value distributions, any distribution learner can be applied, but we prioritize high fidelity and efficiency. VAE often assigns excessive probability mass to less important regions by covering mode [1,2], leading to inaccuracies, especially in complex or multimodal Q-value distributions. GANs face instability and training difficulties due to their adversarial nature and the Nash equilibrium is not easy to reach, and GAN often resulting in mode collapse [3,4]. Thus, we prefer the consistency model for its stability and effectiveness.
> 2.	Please refer to the 1. of the “author rebuttal” at the top for the reason and rationale for estimating uncertainty by using the Q-value distribution of the behavior policy. Moreover, while the estimated uncertainty influences the Q target penalty in the learning policy, it only affects the Q-value in the OOD region and does not impact the updating or optimization of the learning policy in the in-distribution region. Additionally, the uncertainty term has minimal effect on our theoretical proofs; for instance, in the proof of Theorem 4.3 (Appendix D.2), the uncertainty term is ultimately canceled out (see Eq. D.9 to Eq. D.10).
> 3.	Thank you very much for your reminder, we apologize for our oversight and we will take your suggestion to add [5] and [6] to our related work.
>
> About limitation:
>
> We apologize that we have not been very clear about QDQ's limitation. Uncertainty estimation is challenging, but high-precision uncertainty estimation is crucial for the success of Q constraint kind method in offline RL [7]. We hope QDQ can serve as an uncertainty-based RL algorithm to facilitate the successful implementation Q value constraint method in offline RL. However, QDQ has limitations, particularly in using estimated uncertainty to adjust the learning policy's Q target value. Currently, QDQ employs a pessimistic adjustment by dividing the uncertainty value for OOD actions, but this approach may not be optimal. We will continue researching how to combine uncertainty estimation with the LCB of the learning policy, providing theoretical and experimental insights. We apologize for omitting this limitation and will add it to section 7.
>
> [1] Ghasemipour et al. EMaQ: Expected-Max Q-Learning Operator for Simple Yet Effective Offline and Online RL.
>
> [2] Chen et al. OFFLINE REINFORCEMENT LEARNING VIA HIGHFIDELITY GENERATIVE BEHAVIOR MODELING.
>
> [3] Martin et al. Wasserstein generative adversarial networks.
>
> [4] Li et al. MMD GAN: Towards deeper understanding of moment matching network.
>
> [5] Bai et al. Pessimistic Bootstrapping for Uncertainty-Driven Offline Reinforcement Learning.
>
> [6] Mao et al. Supported value regularization for offline reinforcement learning.
>
> [7] Levine et al. Offline reinforcement learning: Tutorial, review, and perspectives on open problems.

---

> > ### Comment · Reviewer_T4W8 · 2024-08-09
> > **Follow-up Questions.**
> >
> > Thank you very much for your response. Most of my questions have been answered, especially as you pointed out, replacing the uncertainty of \(Q\) with that of \(Q^{\beta}\) does indeed seem reasonable. However, I still need to confirm a few points with you. Specifically, do Theorems 4.2 and 4.3 strictly require that \(V(X|(s',a')\) be the uncertainty of \(Q\), or is it permissible to replace it with the uncertainty of \(Q^{\beta}\) in a rigorous sense?

---

> > > ### Author Response · Authors · 2024-08-09
> > >
> > > Thank you very much for your quick response. In QDQ, the uncertainty estimation of $H_Q(a'|s') =  \sqrt{V(X_\epsilon|(s',a'))}$ is only used to penalize the Q-value of actions with high uncertainty($a' \in U(Q)$). Its primary purpose is to make the Q target more pessimistic by adjusting the Q target value to $\frac{1}{H_Q(a'|s')}Q(s',a')1_{(a' \in U(Q))}$. The use of $H_Q(a'|s') =  \sqrt{V(X_\epsilon|(s',a'))}$ does not affect the proofs and conclusions of Theorems 4.2 and 4.3, as it only serves to underestimate the Q target value in high uncertainty (OOD) regions. As long as the Q target value in OOD regions is pessimistic, the theorem holds. Indeed, $H_Q(a'|s') =  \sqrt{V(X_\epsilon|(s',a'))}$ is just a penalty factor, and as long as $\frac{1}{H_Q(a'|s')}Q(s',a')1_{(a' \in U(Q))} < Q(s',a') 1_{(a' \in U(Q))}$, Theorems 4.2 and 4.3 are hold, regardless of whether it is the uncertainty estimation of the Q value of the learning policy or the behavior policy. As shown in Figure G.4 (Appendix G.4), the distribution of $H_Q(a'|s') =  \sqrt{V(X_\epsilon|(s',a'))}$ confirms that $\frac{1}{H_Q(a'|s')}Q(s',a')1_{(a' \in U(Q))} < Q(s',a') 1_{(a' \in U(Q))}$ always holds because the value of $H_Q(a'|s') =  \sqrt{V(X_\epsilon|(s',a'))}$ is relatively large in the high uncertainty regions defined by QDQ.

---

> > > > ### Comment · Reviewer_T4W8 · 2024-08-09
> > > > **Follow-up response**
> > > >
> > > > Thank you for your reply, my doubts have been mostly resolved. I have already improved my score.

---

> > > > > ### Author Response · Authors · 2024-08-10
> > > > >
> > > > > Thank you very much for recognizing our work. We are also deeply grateful for your valuable suggestions, which have helped to further refine and strengthen our research. We will carefully incorporate the points mentioned in the rebuttal and discussion into the paper to make our work clearer and more comprehensive.

---

### Official Review · Reviewer_4FB7 · 2024-07-11

**Soundness:** 3
**Presentation:** 3
**Contribution:** 3
**Rating:** 6
**Confidence:** 4

**Summary:**

This paper proposes a new offline RL method, Q-Distribution guided Q-learning (QDQ), which uses a consistency model to model the distribution of Q-value for uncertainty estimation and then introduces an uncertainty-aware optimization objective for pessimistic Q-learning. This method has theoretical guarantees and exhibits strong performance in the D4RL benchmark.

**Strengths:**

- This paper is clearly written, allowing readers to follow the main arguments.
- It introduces a novel approach for estimating uncertainty of Q-value via a consistency model.
- It provides detailed training curves in the appendix, making the experimental results credible.

**Weaknesses:**

- Personally, there are some confusions regarding the method. The consistency model is trained to fit the distribution of $Q^{\pi{\beta}}$, but it is used to estimate the uncertainty of the updating Q function (as shown in Eq. 8). So, I wonder why the variance of $Q^{\pi{\beta}}$ can estimate the uncertainty of Q. Another confusion is why there is still a policy constraint in Eq. 9. Normally, for pessimistic value offline RL methods like CQL[1], EDAC[2], and PBRL[3], they only impose conservatism on Q-value learning and do not add any constraint on policy learning. I noticed the ablation study on $\gamma$ shows that too small $\gamma$ leads to poor performance on some tasks like hopper-medium-v2. Does this mean that the proposed uncertainty-aware learning objective (Eq. 7) is too weak for some tasks?
- I think some baselines are missing in the main evaluation experiment, like EDAC and PBRL. To the best of my knowledge, EDAC is the SOTA model-free uncertainty-based method so far. So, I suggest the authors include these two algorithms in Table 1 & 2.
- The authors point out that previous uncertainty-based offline RL methods like bootstrap or ensemble impose significant computational burdens (line 181). Although their proposed method does not need ensemble Q-value networks, it additionally brings a consistency model whose training relies on a pretrained diffusion model. So, my concern is whether this will cost more computation compared with ensemble Q-value networks.

[1] Kumar et al. "Conservative Q-Learning for Offline Reinforcement Learning"

[2] An et al. "Uncertainty-Based Offline Reinforcement Learning with Diversified Q-Ensemble"

[3] Bai et al. "Pessimistic Bootstrapping for Uncertainty-Driven Offline Reinforcement Learning"

**Questions:**

To summarize the main points/questions raised in the weaknesses section:

- Could the authors provide some explanations for the confusions mentioned in the first point of weaknesses?
- Could the authors include EDAC and PBRL in Table 1 & 2?
- Could the authors compare the computational costs (like runtime and GPU memory usage) of their method with other methods?

---

> ### Author Rebuttal · Authors · 2024-08-07
>
> For weakness:
>
> 1.	We apologize for the ambiguity. Please see 1. of the “author rebuttal” at the top for the rationale behind using Q-value data from the behavior policy. Although estimated uncertainty penalizes the Q target of the learning policy, it only pessimistic the Q-value in the OOD region and does not affect the learning policy's Q-value updates and optimization in the in-distribution region. We will provide a clearer explanation in Section 3.1.
>
> 2. The gamma term in Eq. 9 stabilizes the learning of a simple Gaussian policy, particularly for action-sensitive and narrower distribution tasks like hopper-medium etc. For these tasks, a simple Gaussian policy can easily sample risky actions as it fits only a single-mode policy. Indeed, the inclusion of a gamma term in QDQ depends on the task. For instance, the optimal gamma for QDQ on wide distribution data (e.g., halfcheetah-medium) is 0. Estimating uncertainty is inherently challenging. QDQ aims to estimate uncertainty directly while applying minimal control over the Q-value and using a less robust Gaussian policy. This combination increases the risk of unstable Q-value training. In fact, the gamma in Eq. 9 is very small relative to the Q-value, primarily to stabilize training and avoid instability in Gaussian policy action sampling. CQL [1], although not explicitly controlling the policy, requires the leaning the behavior policy's distribution and another policy to cover the action space in the dataset, this is some implicit action intervene. IQL [2] uses AWR policy loss to control the learning policy as well as the expectile regression to underestimate the Q value function, and PBRL [3] mentions in the penultimate paragraph of the introduction that simply using ensemble to estimate uncertainty to control Q-value is not very useful, and highlights the need for OOD data in the learning policy's action space to the success of Q value ensemble kinds methods. MCQ follows a similar approach like PBRL and not just constrain the Q value function by behavior policy’s support. To verify the impact of the uncertainty-aware Q-value optimization in QDQ, we compared the performance of Q-values without uncertainty control (Bellman optimization for the Q value function like in online RL setting) to the QDQ algorithm on hopper-medium data, using identical gamma terms setting. Figure 2 in the attached PDF shows the learning curve. The figure illustrates that introducing uncertainty-based constraint for the Q value function in QDQ significantly improves training stability, convergence speed, and learning strategy performance. This supports the effectiveness of QDQ's uncertainty-aware Q-value optimization. We believe stronger learning policy will further reduce the need for this stabilizing term.
> 3.	Thank you for your reminder. We apologize for the oversight and will add EDAC[4] and PBRL[3] to our baseline(Table 2) and related work.
> 4.	Regarding the training cost of the consistency model, we believe it is almost negligible. Training a diffusion model on a 4090 GPU takes about 5.2 minutes, while training a consistency model with this pretrained diffusion model takes about 16 minutes. Additionally, the consistency model is stored and can be reused for subsequent RL experiments, eliminating the need for retraining.
> For the comparison of computational costs, see Table 1 in the “author rebuttal” at the top. The results of other methods are taken from Table 3 of EDAC [4]. QDQ aims not only to achieve SOTA performance but also to fill gaps in uncertainty research and advance methods in Q-value constraints.
>
> For Questions:
>
> Please refer to our previous reply on weaknesses.
>
> Table 2: Comparison of QDQ and the other baselines on the three Gym-MuJoCo tasks. All the experiment are performed on the MuJoCo "-v2" dataset. The results are calculated  over 5 random seeds.med = medium, r = replay, e = expert, ha = halfcheetah, wa = walker2d, ho=hopper
>
> | Dataset  | BC    | AWAC  | DT    | TD3+BC | CQL   | IQL   | UWAC  | MCQ   | EDAC        | PBRL  | QDQ(Ours)   |
> |----------|-------|-------|-------|--------|-------|-------|-------|-------|-------------|-------|-------------|
> | ha-med   | 42.6  | 43.5  | 42.6  | 48.3   | 44.0  | 47.4  | 42.2  | 64.3  | 65.9        | 57.9  | **74.1**  |
> | ho-med   | 52.9  | 57.0  | 67.6  | 59.3   | 58.5  | 66.2  | 50.9  | 78.4  | **101.6** | 75.3  | **99.0**  |
> | wa-med   | 75.3  | 72.4  | 74.0  | 83.7   | 72.5  | 78.3  | 75.4  | 91.0  | **92.5**  | 89.6  | 86.9        |
> | ha-med-r | 36.6  | 40.5  | 36.6  | 44.6   | 45.5  | 44.2  | 35.9  | 56.8  | 61.3        | 45.1  | **63.7**  |
> | ho-med-r | 18.1  | 37.2  | 82.7  | 60.9   | 95.0  | 94.7  | 25.3  | 101.6 | 101.0       | 100.6 | **102.4** |
> | wa-med-r | 26.0  | 27.0  | 66.6  | 81.8   | 77.2  | 73.8  | 23.6  | 91.3  | 87.1        | 77.7  | **93.2**  |
> | ha-med-e | 55.2  | 42.8  | 86.8  | 90.7   | 91.6  | 86.7  | 42.7  | 87.5  | **106.3** | 92.3  | 99.3        |
> | ho-med-e | 52.5  | 55.8  | 107.6 | 98.0   | 105.4 | 91.5  | 44.9  | 112.3 | 110.7       | 110.8 | **113.5** |
> | wa-med-e | 107.5 | 74.5  | 108.1 | 110.1  | 108.8 | 109.6 | 96.5  | 114.2 | 114.7       | 110.1 | **115.9** |
> | Total    | 466.7 | 450.7 | 672.6 | 684.6  | 677.4 | 698.5 | 437.4 | 797.4 | 841.1       | 759.4 | **848.0** |
>
> [1] Kumar et al. "Conservative Q-Learning for Offline Reinforcement Learning".
>
> [2] Kostrikov et al. Offline reinforcement learning with implicit q-learning.
>
> [3] Bai et al. Pessimistic Bootstrapping for Uncertainty-Driven Offline Reinforcement Learning.
>
> [4] An et al. Uncertainty-Based Offline Reinforcement Learning with Diversified Q-Ensemble.

---

> > ### Comment · Reviewer_4FB7 · 2024-08-10
> >
> > Thank you for your reply, my concerns have been mostly resolved. I have already improved my score.

---

> > > ### Author Response · Authors · 2024-08-10
> > >
> > > Thank you very much for your prompt response and for raising the score! Your questions and valuable suggestions play a crucial role in improving our paper, and we will incorporate all the content from the rebuttal into the manuscript. Once again, we sincerely appreciate your feedback and your assistance in enhancing the quality of our paper!

---

### Official Review · Reviewer_rDnz · 2024-07-11

**Soundness:** 2
**Presentation:** 2
**Contribution:** 3
**Rating:** 7
**Confidence:** 3

**Summary:**

The paper proposes a method for estimating Q values in the offline/batch setting by leveraging consistency models. Via these, the uncertainty over the Q function can be estimated and used as a robust penalty to prevent distribution shift in offline RL. The authors provide both experimental validation on the standard D4RL dataset, as well as theoretical insight into the performance of their algorithm.

**Strengths:**

The paper proposes an intuitive and well supported idea to improve the robustness of offline RL. Following in the well established path of using uncertainty instead of pessimism to regularize Q value estimation in the offline regime, they present a solid approach to estimate this uncertainty using consistency models.

As far as I can tell, the addition and evaluation of the consistency model as the measure of uncertainty in a Q function is novel and a solid contribution to the literature.

The empirical results place the method at the top of comparable methods.

**Weaknesses:**

The main weakness at the moment lies in the slightly confusing presentation. While I think that all the ideas are in principle well supported, it is very hard to follow the exact setup and intuition throughout. In addition, I am not fully convinced that the theoretical results are fully informative for the empirical implementation.

The introduction of the consistency model is very terse. For readers who are less deep in the diffusion literature, this presents a large barrier for understanding the rest of the paper. I would recommend that the authors provide a slightly more thorough introduction here.

The sliding window approach to Q estimation seems to rely heavily on a relatively dense informative reward. This should be discussed in the paper. While Theorem 4.1 does (somewhat trivially) hold, it does not suggest that partial reward sums will necessarily be informative. In cases with 0 reward across a chunk (which would happen frequently with sparse rewards) the consistency model would always predict near 0 uncertainty. This issues is currently the main reason I am leaning towards recommending rejection and I am happy to discuss it in the rebuttal in case I am overestimating the importance of the problem.

In general, the theoretical analysis is not tied strongly to the rest of the paper. It is not clear if the authors draw conclusions from it for their method, or simply present it as a justification for its validity (in which case I would expect a slightly more thorough discussion).

The method requires 3 additional hyperparameters, which the authors discuss. Given the enormous difficulty of model selection / hyperparameter tuning in offline RL and the communities lack of coherent standards here, I would like for the authors to discuss how the hyperparameters were tuned (and whether this constitutes test set training). I do concede that this is a wider issue in the community and that I cannot fully fault the authors to adhering to standards here, but I think the issue should be discussed in offline RL papers.

The empirical results do not seem vastly better than MCQ, so i would tone done the writing a small bit. It is unclear whether the authors for example allowed themselves a larger hyperparameter tuning budget.

**Questions:**

Why is the set of baseline methods different on the two different benchmarks? Especially the strongest method on D4RL seems to be missing from the Ant-maze comparison.

---

> ### Author Rebuttal · Authors · 2024-08-07
>
> For weakness:
>
> Firstly, we apologize for any ambiguities. In offline RL, which aims to train policy without interacting with the environment, "distribution shift" is the main obstacle. A learning policy may take out-of-distribution (OOD) actions, leading to overestimated Q-values. One suggested way to tackle this overestimation in Q value function is by its high uncertainty property in the OOD regions [3]. Following this idea, QDQ targets high uncertainty of the Q-values in OOD regions. Please see more details in the motivation of the "author rebuttal" at the top.
>
> While our theoretical framework is detailed in Section 4, we reference these results in Section 3 to provide context. For instance, Theorem 4.1, mentioned in line 166 of Section 3.1, shows that our sliding window-based truncated Q-value distribution converges to the true Q-value distribution, which guarantee accurate uncertainty estimation. Each theoretical result is briefly introduced in Section 4 before being detailed, such as in lines 232-235, where Theorem 4.2 is discussed. This theorem shows that the consistency model is suitable for uncertainty estimation because it guarantees the effect of actions on the variance change of the Q-value. Theorems 4.3 and 4.4 illustrate that QDQ penalizes the OOD region by uncertainty while ensuring that the Q value function in the in-distribution region is close to the optimal Q-value, which is the goal of offline RL. Detailed descriptions, proofs, and implications of these theorems are provided in Appendices B-F. We will revise the statements further for clarity.
>
> 1.	For the consistency model: we use the consistency model [1] for the Q-value distribution learner, which addresses the inconsistency between the diffusion model's prior information and the target distribution. It also improves sampling speed through one-step sampling, outperforming the diffusion model. In essence, the consistency model is an enhanced generative model compared to the diffusion model. The diffusion model gradually adds noise to transform the target distribution into a Gaussian distribution and by estimating the random noise to achieve the reverse process, i.e., sampling a priori samples from a Gaussian distribution (forms the sample generation trajectory) and denoise to the target sample. And the consistency model ensures each step in a sample generation trajectory of the diffusion process aligns with the target sample (we call consistency), enhancing uncertainty estimation. The consistency feature, as discussed in Theorem 4.2, ensures the accurate impact of action changes on the variance of the final bootstrap samples, making Q-value uncertainty more sensitive to OOD actions compared to the diffusion model. Additionally, the fast-sampling process of the consistency model enhances QDQ's efficiency. Despite some quality loss in restoring real samples, this loss is negligible for QDQ, as it only calculates uncertainty from the variance of the bootstrap sample, not use the absolute Q-value of the sampled samples. Overall, the consistency model is an ideal distribution learner for uncertainty estimation due to its consistency, high fidelity, ease of training, and faster sampling.
> 2.	For the sliding window: in fact, a good Q-value dataset for uncertainty estimation should span a broad state and action space of the dataset to accurately characterize the Q distribution and detect high uncertainty in the OOD action region. This coverage of the state and action allows us to identify actions with high Q-value uncertainty in OOD regions. Even if the variance of Q-values in the in-distribution region is small, it makes uncertainty detection more sensitive, as Q-value functions typically exhibit high uncertainty in OOD regions [3], aiding in identifying OOD actions by comparison. We also give some rational on how to choose the sliding window size, please see the line 814-821 in Appendix G.1. For sparse reward tasks, we apply adjustments such as those in IQL [2] to prevent all rewards from being zero and maintain some level of variance of the bootstrap samples even in the in-distribution region. Although a wider sliding window theoretically provides richer information as suggested by the reviewer, experiments (e.g., Fig. 1 in the attached PDF) show that it has not influence the shape of distribution of the Q-value data much. Conversely, larger window width reduces the Q-value dataset size, affecting coverage of the Q-value dataset, then harm the uncertainty estimation accuracy.
> 3.	For parameter tuning, please refer to 2. in the” author rebuttal” at the top.
> 4.	We have updated our results during parameter tuning, as shown in Table 2. QDQ outperforms MCQ by 50 points, with notable improvements in tasks like HalfCheetah-Medium and Hopper-Medium. MCQ addresses Q-value overestimation by identifying in-distribution and OOD actions from an estimated behavior policy. While both QDQ and MCQ apply mildly constraints on the Q value, they follow different approaches. QDQ focuses on Q-value constraints introduced in [3], whereas MCQ is more aligned with the method used in policy control method. Our goal with QDQ is not merely to achieve SOTA results but to advance research on uncertainty-guided pessimistic adjustment of Q-values in offline RL. QDQ provides both theoretical support and experimental validation, aiming to improve offline RL methodology and address research gaps beyond SOTA performance.
>
>
> For the question:
>
> In footnote 1, we explain why we did not include the same baselines as in Table 1 of Section 5, specifically the results of UWAC and MCQ. They do not provide experiment results for the Antmaze task, and we lack information on how to set their hyperparameters.
>
> [1] Song et al. Consistency models.
>
> [2] Kostrikov et al. Offline reinforcement learning with implicit q-learning.
>
> [3] Levine et al. Offline reinforcement learning: Tutorial, review, and perspectives on open problems.

---

> > ### Comment · Reviewer_rDnz · 2024-08-11
> > **Reviewer reply**
> >
> > Thanks for your thorough comments. I am satisfied with the answer and I will update my score, but I truly believe that this paper deserves another very thorough editorial pass to make everything more clear to the readers. I do not mean to offend, but the presentation feels very rushed and it might be a good idea to consider re-submission to a future conference after polishing the presentation so that you do not lose potential impact due to low clarity. As I am only a reviewer, this is of course not my final decision and between the AC/SAC and the authors, but I think it might be a good idea to forgo publication now to give yourselves the time to make your work truly shine.

---

> > > ### Author Response · Authors · 2024-08-12
> > >
> > > Thank you very much for your response! We sincerely appreciate your ongoing recognition of our work and its contribution to the field of offline RL. As we mentioned earlier, we hope that QDQ can provide evidence and insights for research on uncertainty estimation in offline reinforcement learning, and on how to effectively use uncertainty to more accurately control the overestimation bias of Q values in areas beyond the knowledge of the dataset. We hope our work can offer some insights and support for future research in offline RL, and even more, shed a light on the usage of uncertainty estimation in the study of exploration in online RL.
> > >
> > > We sincerely apologize for the issues with the presentation of our work, which negatively impacted your review experience and took up your valuable time. We are taking this feedback very seriously and will thoroughly revise all presentation-related aspects of the paper. In fact, we have already begun revising our manuscript, and we will incorporate all the issues you mentioned, as well as the content from the rebuttal, into the latest version of our paper. We will refine the motivation, the introduction of relevant background (especially the consistency model), the description of the algorithm, and particularly the integration of theory and algorithm. We are confident that, with the additional time available and the valuable suggestions provided by you and the other reviewers, we can complete these revisions effectively. We are committed to refining our paper and are confident that the next version will meet your expectations.
> > >
> > > Once again, thank you for recognizing our work, and we sincerely appreciate your review efforts! We will address and correct all the issues you mentioned, making the entire work clearer and more precise. We also hope that our work can meet your expectations and play a positive role in advancing research in reinforcement learning.

---

> > > > ### Comment · Reviewer_rDnz · 2024-08-12
> > > > **Quick reply**
> > > >
> > > > If you are confident that you can get the work in shape for a great camera ready than I’m not going to stand in the way of acceptance :) ultimately it’s the AC/SAC/PCs that decide of course, I just wanted to make sure that you are certain you can get this work into a great enough shape so that it can get the recognition it deserves

---

> > > > > ### Author Response · Authors · 2024-08-13
> > > > >
> > > > > Thank you very much for your response! We sincerely appreciate you pointing out our issues and placing your trust in us. We will do our utmost to polish and revise the paper, ensuring that we live up to your trust and expectations! Regardless of the outcome, we will carefully implement your suggestions in our revisions and will continue to adhere to the rigorous scholarship that you and the other reviewers have demonstrated. Once again, thank you for your time and effort!

---

### Author Rebuttal · Authors · 2024-08-07

We appreciate the valuable comments from our three reviewers, which have helped us improve our manuscript. We have provided detailed answers to each question and included additional experiments in the attached PDF. To address any remaining confusion, we would like to introduce the motivation behind our work (QDQ).

QDQ aims to directly control overestimation in offline Reinforcement Learning (RL) for Q value functions by estimating the uncertainty of Q-values concerning the action during training [1]. Estimating Q-value uncertainty is challenging [1], and to our best knowledge, most methods address it indirectly by using ensembles of Q-value functions rather than bootstrapping. These ensemble methods suffer from lacking diversity in the Q-value[2] and fail to accurately represent the true Q-value distribution, often requiring tens or hundreds of Q-values to improve accuracy, which is computationally inefficient [2,4]. Other methods, like CQL[6] and MCQ, focus on underestimating Q-values in the OOD region by first identifying the OOD region, rather than exploiting Q-value uncertainty in that region.

QDQ aims to solve the problem of estimating Q-value uncertainty by directly computing this uncertainty through bootstrap sampling from the distribution of Q-values of the behavior policy. By approximating the behavior policy's Q-values from the dataset, we train a high-fidelity, efficient distribution learner-consistency model. This ensures the quality of the learned Q-value distribution. Since the behavior and learning policy share the same high uncertainty action set [4], we can sample from the learned Q-value distribution to estimate uncertainty, identify risky actions, and make Q target values for these actions more pessimistic. Additionally, QDQ proposes an uncertainty-aware Q-value optimization objective to avoid excessively penalizing Q-values, ensuring the Q value function’s exploratory in the in-distribution region. QDQ seeks to find the optimal Q-value that exceeds the behavioral policy's optimal Q-value while being as pessimistic as possible in the OOD region.

QDQ aims not only to achieve state-of-the-art experimental results but also to explore and advance uncertainty estimation in RL, an area with limited research. By promoting the role of uncertainty estimation in offline RL, QDQ seeks to enhance the methodology completeness of Q-value constraint. It is designed to be concise while maintaining competitive experimental performance and efficiency (Table 1). These features make QDQ flexible for integration into more complex elements, such as, enabling the incorporation of more powerful policies and enhancing exploration in online RL [1].

Next, we provide a brief description of some confusing details for QDQs:

1.	We chose to estimate the Q-value distribution of the behavior policy rather than the learning policy because they share almost the same high-uncertainty action set [4]. Using the behavior policy’s Q-value distribution offers several advantages. Firstly, the behavior policy's Q-value dataset is derived from the true dataset, ensuring high-quality distribution learning. Conversely, the learning policy's Q-value is unknown, counterfactually learned, and often noisy and biased. Poor data quality would lead to biased distribution learning. Secondly, using the behavior policy's Q-value distribution to identify high-uncertainty actions does not constrain the learning policy's target Q-value to match the behavior policy's. This aligns with our goal of a mildly constrained Q-value. Theorems 4.3 and 4.4 demonstrate that our uncertainty-aware Q-value optimization objective can train a Q-value that closely approximates the optimal Q-value in the in-distribution region, outperforming the behavior policy, as confirmed by our experimental results. We will integrate these points into the paper to clarify the QDQ algorithm.
2.	Although QDQ has three hyperparameters ($\alpha$, $\beta$, and $\gamma$) to achieve more flexible functions, the tuning process is straightforward. Take $\alpha$ as example, as discussed in Theorem 4.4 (Appendix E), theoretically $(1-\alpha)(1-\beta)$ should be small. Since beta controls the size of the uncertainty set and needs flexibility across different tasks, we typically set $\alpha$ closer to 1, tuning it between 0.9 and 0.995, which requires only a few experiments to find the optimal value. Our tuning process involves sequentially fixing parameters while choosing the best $\alpha$, then $\beta$, and finally $\gamma$. Considering the characteristics of different datasets (Section 5.2 and Appendix G.3), we can set each parameter a confidence initial value that close to its optimal value. QDQ provides evidence-based guidelines for hyperparameter ranges, making tuning manageable. Additionally, QDQ's speed (Table 1) reduces the tuning burden. This analysis as well as the tuning detail of $\beta$ and $\gamma$ will be detailed in Section 5 and Appendix G.3.

Table 1: Computational performance of QDQ and other SOTA methods

|           | Runtime(s/epoch) | GPU Mem.(GB) |
|-----------|------------------|--------------|
| SAC       | 21.4             | 1.3          |
| CQL       | 38.2             | 1.4          |
| EDAC      | 30.8             | 1.8          |
| **QDQ** | **0.028**      | **0.74**   |

[1] Levine et al. Offline reinforcement learning: Tutorial, review, and perspectives on open problems.

[2] An et al. Uncertainty-Based Offline Reinforcement Learning with Diversified Q-Ensemble.

[3] Agarwal et al. An Optimistic Perspective on Offline Reinforcement Learning.

[4] Kumar et al. Stabilizing off-policy Q-learning via bootstrapping error reduction.

[5] Kostrikov et al. Offline reinforcement learning with implicit q-learning.

[6] Kumar et al. "Conservative Q-Learning for Offline Reinforcement Learning".

---

### Decision · Program_Chairs · 2024-09-25

**Decision:**

Accept (poster)

**Comment:**

The paper presents a method for conservative offline RL via consistency models, that reviewers found well-motivated, intuitive, novel, and sound. The paper itself is mostly clear and the empirical evaluation is satisfactory. Some concerns were raised regarding the presentation and correctness of the method's theoretical grounding, as well as its context in the literature. The reviewers and I are satisfied with the authors' explanation of these issues and with their promise to polish the writing.